# Discovery and Biotechnological Exploitation of Glycoside-Phosphorylases

**DOI:** 10.3390/ijms23063043

**Published:** 2022-03-11

**Authors:** Ao Li, Mounir Benkoulouche, Simon Ladeveze, Julien Durand, Gianluca Cioci, Elisabeth Laville, Gabrielle Potocki-Veronese

**Affiliations:** Toulouse Biotechnology Institute (TBI), Centre National de la Recherche Scientifique (CNRS) Toulouse Biotechnology Institute (TBI), French National Research Institute for Agriculture, Food and Environment (INRAE), Institut National des Sciences Appliquées (INSA), Université de Toulouse, 31000 Toulouse, France; liao13343445081@hotmail.com (A.L.); mounir.benkoulouche@gmail.com (M.B.); simon.ladeveze@insa-toulouse.frr (S.L.); julien.durand@insa-toulouse.fr (J.D.); cioci@insa-toulouse.fr (G.C.); laville@insa-toulouse.fr (E.L.)

**Keywords:** glycoside phosphorylases, carbohydrate-active enzymes, glycosides, glycochemistry, biotechnology, functional metagenomics, sequence similarity networks, screening

## Abstract

Among carbohydrate active enzymes, glycoside phosphorylases (GPs) are valuable catalysts for white biotechnologies, due to their exquisite capacity to efficiently re-modulate oligo- and poly-saccharides, without the need for costly activated sugars as substrates. The reversibility of the phosphorolysis reaction, indeed, makes them attractive tools for glycodiversification. However, discovery of new GP functions is hindered by the difficulty in identifying them in sequence databases, and, rather, relies on extensive and tedious biochemical characterization studies. Nevertheless, recent advances in automated tools have led to major improvements in GP mining, activity predictions, and functional screening. Implementation of GPs into innovative in vitro and in cellulo bioproduction strategies has also made substantial advances. Herein, we propose to discuss the latest developments in the strategies employed to efficiently discover GPs and make the best use of their exceptional catalytic properties for glycoside bioproduction.

## 1. Introduction

As numerous glycosides are used in the food, feed, cosmetics, health, and chemical industries, glycoside-synthesizing enzymes are very interesting tools to produce these compounds in vitro or in cellulo [1,2]. Indeed, while today most glycosides are chemically synthesized, biocatalysts can bypass or be used in combination with glycochemistry [3,4]. There are four types of glycoside-synthesizing enzymes.

In cellulo, glycosidic bond formation is mainly performed by glycosyl transferases (GTs), which catalyze the transfer of sugar moieties from an activated donor onto an acceptor molecule (often a carbohydrate) [5]. The advantages of GTs are the high synthesis yield they often offer [6]; while, major drawbacks are (i) the requirement for expensive donors [7], which can, nevertheless, be produced in cellulo, together with the target GTs to directly produce and excrete oligosaccharides [8,9], and (ii) the low number of stable GTs, produced as soluble forms for in vitro synthesis purposes [10].

Transglycosidases (TGs) catalyze the transfer of a glycosyl unit from a donor glycoside (often readily available and cheap, unlike nucleotide sugars) onto an acceptor molecule, to produce another type of carbohydrate or glycoconjugate, thanks to their wide acceptor promiscuity. More and more attention is being given to the discovery of new TGs [11] and their engineering [12,13,14,15], for oligosaccharide synthesis in vitro. However, only a few native TGs have been described, and their specificities are restricted to some substrates.

Glycosynthases (GSs) are engineered glycoside hydrolases (GHs), in which the substitution of the catalytic nucleophile by a non-nucleophilic residue (alanine, glycine, or serine) abolishes the innate hydrolytic activity. The role of the nucleophile is played by an activated donor, such as fluorine-containing substrates, providing the energy necessary to cross the barrier for the reaction to occur in the synthesis direction. They can, therefore, synthesize glycosidic bonds instead of hydrolyzing them. However, fluorine donor substrates are expensive or have to be prepared for each study, and suffer from instability [16,17]. Replacing them with sugar oxazoline [18] or (more toxic) glycosyl azide [19] donors could result in a substantial increase of synthesis yields [20].

Glycoside phosphorylases (GPs) catalyze, both glycoside degradation by using phosphate to breakdown osidic linkages (phosphorolysis), and synthesis by using sugar-phosphates as glycosyl donors (reverse phosphorolysis). GPs were discovered by the 1947 physiology and medicine Nobel Prize recipients Carl F. and Gerty T. Cori [21], who greatly contributed to the understanding of glycogen metabolism. Initially the term ‘phosphorylases’ was used to describe the enzyme found in the liver and skeletal muscle fibers that can produce α-d-glucose-1-phosphate (αGlc1P) from glycogen. GPs share structural and mechanistic characteristics with both glycoside hydrolases and glycosyl transferases. The two reactions of phosphorolysis and reverse phosphorolysis catalyzed by GPs oppose one another, resulting in an equilibrium. This is due to the fact that the free energy required for the cleavage of the glycosidic bond is close to the one required for cleavage of the ester linkage in glycosyl phosphates, as demonstrated by a study on cellobiose phosphorylase [22]. Herein, we propose to discuss these enzymes, with a particular focus on the latest advances in the approaches employed for their discovery and characterization. We will also emphasize the most up to date developments regarding the industrial applications of these particularly attractive biochemical tools, including in synthetic biology.

## 2. Glycoside-Phosphorylases

### 2.1. GP Catalytic Mechanism

Owing to the chimeric nature of GPs relative to GHs and GTs, they can be found in both families. A list of biochemically characterized GPs, as of November 2021, can be found in Table 1. GPs can be classified into retaining GPs or inverting GPs, based on the change in the anomeric carbon configuration after the reaction [23]. Km values span across two orders of magnitude, from 0.3 [24] to 9.6 mM [25], 1,46 [24] to 130 mM [26], and 0.42 [24] to 7.1 mM [27], for donors, acceptors and phosphate, respectively. Turnover numbers also widely vary. Kcat values can be as low as 0.8 s^−1^ for GH65 glucosylglycerate phosphorylase [28], up to 2866 s^−1^ for GH94 cellodextrin phosphorylase on cellopentaose [29].

Both inverting GHs and GPs follow a direct displacement, SN_2_-like reaction mechanism, (Figure 1) but differ in the catalytic machinery employed to perform the reactions [175]. In the case of inverting GHs, two catalytic amino acid residues are required: a proton donor that performs the nucleophilic attack on the anomeric carbon, and a catalytic base which activates a water molecule. Inverting GPs, on the contrary, require a single catalytic residue, since the catalytic base is the inorganic phosphate itself (Pi). The anomeric configuration inversion is achieved in a single step; the sugar at the non-reductive end is attacked by the catalytic base at the C_1_ position, a proton provided by the acid/base residue is then captured by the inter-osidic oxygen, which leads to the formation of an oxocarbenium ion-like transition state. Bond cleavage completes the reaction, yielding a glycosyl-phosphate of inverted configuration and an oligosaccharide of reduced chain length. Most retaining GPs (Figure 2) act on α-glycosides. Only a few GH3 family members are reported to be retaining GPs that convert β-glycosides to β-glycosidic phosphate [31]. The mechanism adopted by these enzymes is close to that of retaining GTs. They can either form a glycosyl-enzyme intermediate via double displacement mechanism, or operate through front-side displacement mechanism [175] (Figure 1). The double displacement mechanism occurs in two steps: First, the nucleophile attacks the C_1_ of the sugar at the non-reductive end, concomitantly to a proton capture by the inter-osidic oxygen on the proton donor. The inter-osidic oxygen-C_1_ bond cleavage results in the formation of a glycosyl-enzyme intermediate, plus a glycoside of reduced chain length. Following the withdrawal of this product from the catalytic site, the proton donor retrieves a proton from the inorganic phosphate. This activated inorganic phosphate participates in a concomitant nucleophilic attack onto the C_1_ of the glycosyl-enzyme intermediate. The reaction concludes with the cleavage of the glycosyl–enzyme bond, yielding a glycoside–phosphate of retained configuration (Figure 1).

The front-side displacement mechanism (also known as internal return-like or *S*N*i*-like), likely used by GPs of the GT4 and GT35 families, requires a single catalytic amino acid [47]. Only the nucleophile is required, since the proton donor role is played by the inorganic phosphate itself. A hydroxyl from the orthophosphate provides the proton caught by the glycosidic oxygen when a nucleophilic attack occurs. It is worth noting that both the protonation and the nucleophilic attack occur on the same side of the sugar ring, to ensure retention of anomery. The oxocarbenium-ion transition state is stabilized by a nucleophile, namely, in this case, a glutamine or an asparagine.

### 2.2. Classification and Substrate Specificity

In the Carbohydrate-Active Enzymes (CAZy) Database, GPs are classified into a total of 11 CAZymes families (Table 1). It is worth noting that a *N*-acetylglucosaminidase from the GH84 family has recently been converted into an efficient GP by a single point mutation [176]. However, as GH84 enzymes do not include any native GP, this family will not be further mentioned.

#### 2.2.1. Retaining GPs

Retaining GPs are found in two GHs families (GH3 and GH13_18) and two GTs families (GT4 and GT35). Only two GPs have so far been characterized in the large GH3 family, which contains 42,678 members (November 2021). Nag3 from *Cellulomonas fimi,* which was initially characterized as a bifunctional *N*-acetyl-β-glucosaminidase/β-glucosidase [177], was actually found to also be a β-*N*-acetylglucosaminide phosphorylase releasing β-d-*N*-acetyl-glucosamine-1-phosphate (βGlc*N*Ac1P). This new enzymatic activity is due to the presence of a His as a catalytic acid/base residue, instead of the canonical Glu or Asp. The preferred reaction of Nag3 is the cleavage of the disaccharide GlcNAc-anhydro-MurNAc for the recycling of peptidoglycan [31,178]. Another GH3 β-glucoside phosphorylase, BglP, was further discovered by activity-based metagenomics. BglP phosphorolyses the β-1,4-glucosidic linkages of cellulose and cello-oligosaccharides [30]. All the other retaining GPs display an α-linkage specificity, such as the retaining GPs from the GH13_18 subfamily, which act on α-glucosides to produce α-glucose 1-phosphate (αGlc1P).

The GH13_18 subfamily was thought for a long time to contain only sucrose phosphorylases. Nevertheless, sucrose 6′-phosphate phosphorylases (releasing αGlc1P and α-d-fructose-6-phosphate) were recently discovered from *Thermoanaerobacterium thermosaccharolyticum* [48] and *Ruminococcus gnavus* [44]. Later on, a novel glucosylglycerate phosphorylase from *Meiothermus silvanus* was discovered [28]. This enzyme is involved in the metabolism of α-1,2-glucosylglycerate, a molecule known to protect cells from water loss [179]. An α-1,2-glucosylglycerol phosphorylase from *Marinobacter adhaerens* was further identified. This enzyme is thought to help the bacterium tolerate high salt concentrations, thanks to the protective effect of glucosylglycerol [42]. The sucrose phosphorylases and related enzymes from GH13, their applications, and engineering were recently reviewed in [180].

GPs from the GT4 family are all trehalose phosphorylases [118,119,121,181], while all the characterized members of the GT35 family are α-1,4-glucan phosphorylases, acting on glycogen, starch, and maltodextrins [182,183].

#### 2.2.2. Inverting GPs

Inverting GPs are currently grouped into seven GH families (GH65, GH94, GH112, GH130, GH149, GH161) and one GT family harboring dual glycosyltransferase/phosphorylase activities (GT108) [174]. All those of the GH65 family are involved in the cleavage of α-glucosides, to produce β-d-glucose-1-phosphate (βGlc1P). The first GP from this family was shown to effectively convert maltose and phosphate into βGlc1P and d-glucose [54]. Other enzymes with new specificities, playing key roles in various metabolic pathways, were further identified, such as the trehalose-6-phosphate phosphorylase from *Lactococcus lactis subsp. Lactis* [62], the trehalose phosphorylase from *Thermoanaerobacter brockii* [184], the kojibiose phosphorylase from *Thermoanaerobacter brockii* [185], the nigerose phosphorylase and the 3-O-α-d-Glucopyranosyl-l-rhamnose phosphorylase from *Clostridium phytofermentans* [25,65], and the 2-O-α-d-glucosylglycerol phosphorylase from *Bacillus selenitireducens*. The latter utilizes polyols such as glycerol as acceptors rather than longer oligosaccharides or polysaccharides, and could be involved in both the biosynthesis and cleavage of 2-O-α-d-glucosylglycerol, providing the microorganism with adaptation to environments with high salts concentration [67].

Inverting GPs from the GH94 family act on various β-linked glycosides to produce αGlc1P or α-d-*N*-acetyl-glucosamine-1-phosphate (αGlc*N*Ac1P). The cellobiose phosphorylase and cellodextrin phosphorylases from *Clostridium stercorarium* [69] were the first to be reported in this family. Later, a *N,N*′-diacetylchitobiose phosphorylase able to generate α-d-*N*-acetyl-glucosamine-1-phosphate was identified from *Vibrio furnissii* [84]. All the GH94 GPs discovered later on were specific for αGlc1P. They include enzymes acting on various β-glucosidic linkages. Among them, the laminaribiose phosphorylase from *Paenibacillus* sp. [82]) plays a key role in the metabolism of paramylon, an intracellular storage form of β-1,3-glucan usually found in *Euglena* species [186]. The laminaribiose phosphorylase from *Acholeplasma laidlawii* was found to effectively produce shorter 1,3-β-d-glucosyl disaccharides [92]. The fungal (*Neurospora crassa*) and bacterial (*Xanthomonas campestris*) cellobionic acid phosphorylases catalyze the reversible phosphorolysis of 4-O-β-d-glucopyranosyl-d-gluconic acid, a motif found in recalcitrant cellulosic biomass [90]. Finally, the β-1,2-oligoglucan phosphorylase from *Listeria innocua* is hypothesized to participate in the metabolism of sophorose-containing glucans and/or exogenous 1,2-β-glucans in specific environmental conditions [87].

The GH112 family contains inverting phosphorylases acting on β-galactosides, which release α-d-galactose-1-phosphate (αGal1P). β-d-galactopyranosyl-1,3-*N*-acetylhexosamine phosphorylases are subdivided, based on their substrate preference, into galacto-*N*-biose phosphorylase (GNBP) [99,187], lacto-*N*-biose I phosphorylase (LNBP) [95], and galacto-*N*-biose/lacto-*N*-biose I phosphorylase (GLNBP, with no clear preference for either galacto-*N*-biose or lacto-*N*-biose I) [99,188]. The d-galactosyl-β-1,4-l-rhamnose phosphorylase from *Clostridium phytofermentans* probably plays a role in the degradation of rhamnogalacturonan I (RG-I), an important component of pectin [99].

The GH130 family contains inverting phosphorylases that act on β-mannosides to generate α-d-mannose-1-phosphate (αMan1P). The identification of a 4-O-β-d-mannosyl-d-glucose phosphorylase activity led to the creation of the GH130 family [105]. Then, many different substrate specificities were found, highlighting their role in the catabolism of plant, mammal, bacterial, and fungal mannosides, in particular in the human gut, where these bacterial enzymes are abundant [189]. The β-1,4 mannooligosaccharide phosphorylases, first discovered by Kawahara and colleagues [104], and the 4-O-β-d-mannosyl-d-glucose phosphorylases participate in the breakdown of plant cell wall mannans and glucomannans. The β-1,4-mannosyl-*N*-acetyl-glucosamine phosphorylase and β-1,4-mannopyranosyl-chitobiose phosphorylase are both involved in the catabolism of eukaryotic *N*-glycans [108,109]. The β-1,2-mannobiose phosphorylases [100,101] and β-1,2-oligomannan phosphorylases [101] act on exogenous yeast mannan. A β-1,3-mannobiose phosphorylase was later discovered [102], but its physiological function remains unelucidated. In 2013, Ladevèze and colleagues indicated that, based on sequence similarities, the GH130 family could be subdivided into at least two subgroups. The GH130_1 subgroup would include enzymes strictly acting on β-d-Man*p*-1,4-d-Glc; and the GH130_2 one, those acting on β-1,4-mannosides. The authors also mentioned the presence of a large heterogeneous sequence cluster, named GH130_NC, which was further shown to include β-1,2-mannosidases [190] and β-1,3-mannobiose phosphorylases [102]. Li and colleagues [112] revisited the functional diversity of GH130 enzymes, using a sequence similarity network approach to identify 15 iso-functional sequences meta-nodes. Three new enzymes originating from uncharacterized clusters were studied, which led to the identification of a new function, β-d-Man*p*-1,4-d-GlcA phosphorylase. This motif is found in the O-antigen of *Shigella* species lipopolysaccharides [191].

GH149 and GH161 are newly created families, with characterized enzymes originating from the photosynthetic excavate *Euglena gracilis*, the heterokont *Ochromonas* spp., and other bacteria that act on β-1,3-glucans, producing αGlc1P. The members of the GH149 and GH161 families likely evolved from a common ancestor, since they harbor the consensus amino acid characteristic of GP activity. However, as their sequences share only about 20% identity, they were segregated into the distinct GH149 [115] and GH161 families [116].

Four GPs were recently characterized in the newly created GT108 CAZy family. They act on mannogen, a β-1,2-mannan found in Leishmania parasites used for energy storage. Interestingly, these four enzymes are dual glycosyltransferase-phosphorylases, which take part in the recycling of mannogen. These Leishmania enzymes bear the same fold and adopt a similar reaction mechanism as bacterial GH130 mannan phosphorylases or hydrolases. However, their significant difference at the sequence level justified classifying them in another family [174].

Taken together, these studies show that GPs can, thus, degrade and synthesize a large array of glycosidic linkages, with the notable exceptions of α-/β-1,6 and β-1,1 linkages. As of November 2021, only 1893 phosphorylases had been characterized (Table 1), which represents about 1% of all the characterized CAZymes [192,193]. Taking into consideration the huge number of GTs and GHs that are listed in the CAZy database, many GPs with novel specificities might remain to be discovered.

### 2.3. Tertiary and Quaternary Structures

With the exception of GT4 retaining trehalose phosphorylases and GH161 inverting β-1,3 glucan phosphorylases, for which no structure has been elucidated yet, at least one GP crystallographic structure is available for each family in the protein data bank (PDB) (Table 2, Figure 3).

Thanks to these structures, in particular the ones in complex with substrates and products, the catalytic residues have been identified, such as those involved in phosphate binding. These structures complement the site-directed mutagenesis studies that have been performed to identify the critical residues for GP activity (e.g., [109] for the GH130 family). Additionally, when several GP members from the same family have been both functionally and structurally characterized, the structural comparison not only allows identifying the molecular determinants of GP activity but also those establishing donor and acceptor specificities ([189,194] for the GH130 family, for example), or even allows prediction of the phosphorolytic or hydrolytic nature of sequence clusters, together with the linkage specificity of the reactions [112].

From the structural point of view, GPs adopt a range of common 3D-folds, including (α/α)6 and (β/α)_8_ barrels, five-bladed beta-propeller fold, and GT-B folds. GPs containing CAZY families are univocally associated with a common fold, such as GH130 enzymes, which are all five-bladed beta-propellers. For enzymes from different CAZy families, there is no correlation between 3D-folds and GPs mechanism or substrate specificity, except for GPs belonging to the GH130 and GT108 families, which share a common five-bladed beta-propeller fold and act, for some of them, on the same substrates with the same inverting mechanism.

The vast majority of GPs are present in solution as dimers and, as such, are catalytically active, although no clear demonstration of allostery between the two protomers has been evidenced so far. A few of them are monomers or they assemble as larger homooligomeric complexes, such as the hexameric Uhgb_MP enzyme, for which the catalytic pocket results from the assembly of the different monomers (Table 2). However, the multimerization interface and the precise topology of GP active sites remains difficult to predict [195].

### 2.4. Biotechnological Applications

The different approaches for glycoside production include polysaccharide extraction from natural sources (e.g., microorganisms or plants) [227,228], hydrolysis and size-fractionation [229], chemical synthesis [7], (chemo-)enzymatic synthesis [7,230], and whole cell or cell-free synthesis [230,231]. Due to the high diversity of glycosides found in nature, direct extraction from natural sources remains the method of choice, especially for high-scale applications. However, several purification steps are required, often making use of harsh chemicals or mechanical extraction methods. Resource competition between food and feed industries can also appear when they are produced from raw plant material. This method is, thus, restrained to the extraction of abundant glycosides, such as sucrose (for the food industry) or cellulose (paper industry, biofuels) [227,232].

Chemical synthesis is well adapted for the synthesis of both structurally simple and more complex carbohydrates. Owing to its versatility, it is indeed possible to either introduce rare glycosyl units in glycosides or to assemble them in a more modular or complex fashion. Several automated systems were recently investigated [233,234], and this led to the chemical synthesis of complex glycoside libraries [235,236]. While being currently addressed [237], major drawbacks intrinsic to the use of these chemical syntheses are inevitable. One can think about the tedious manipulation of protecting groups leading to multi-step pathways, the use of (often toxic) solvents and harsh temperature conditions, or the lack of reaction selectivity [7]. Even though some carbohydrates cannot be accessed through chemical synthesis or at the cost of low yields, chemical synthesis still remains one of the most used methods to produce specialty and high added-value glycosides at industrial scale.

Finally, enzyme-based approaches, sometimes in combination with chemical steps to take advantage of the best of both worlds, are increasingly being envisioned for the synthesis of glycosides [7,110,238]. Enzymes, indeed, offer exquisite regio- and stereo-specificities, while being able to work under mild pH and temperature conditions in aqueous solvents. Enzymatic glycoside synthesis now benefits from an expanded catalogue of characterized biocatalysts found in many CAZy families, which can even be further engineered, when lacking the requisite properties [2,10]. Whole-cell biocatalysts also received growing interest, especially for the in cellulo coupling of carbohydrates to lipids or proteins, allowing the production of complex glycoconjugates for the pharmaceutical industries [230]. Examples of enzymatic syntheses strategies involving GPs are presented in Figure 4.

As explained above, GPs can be employed for the synthesis of high-value oligosaccharides, polysaccharides, and glycoconjugates, by reverse-phosphorolysis from glycosyl phosphates. Likewise, the phosphorolysis reaction can be used to regenerate glycosyl phosphates for multi-step enzymatic syntheses, for which a review was recently published [238]. Several examples of glycoside structures accessed through enzymatic synthesis using GPs were previously reported and reviewed [175,239,240]. Let us mention the case of the GH130 Uhgb_MP enzyme, which can use the relatively cheap *N*-acetyl-d-glucosamine and αMan1P substrates to produce the highly expensive β-1,4-mannopyranosyl-chitobiose; of major interest as the core part of human *N*-glycans [109]. Crystalline linear β-1,4-mannan, a major component of plant cell-wall hemicelluloses, has been successfully produced using the GH130 β-1,4-mannooligosaccharide phosphorylase TM1225 [110]. Similarly, sucrose-phosphorylases can use a cheap and renewable donor substrate, to synthesize α-1,2-glucosylglycerol, a compound used in cosmetics [241]. As previously mentioned, the ability of GPs to generate sugar phosphates from low-cost substrates by phosphorolysis can advantageously be used in enzymatic cascades, to regenerate nucleotide sugar donors essential to GTs (Figure 4). This was exemplified in vitro, with the kg-scale production of LNB from sucrose and *N*-acetyl-glucosamine, using an elegant one-pot four-enzyme system [239,240], and also in cellulo for the production of phenolic glucosides from sucrose in *Escherichia coli* [242], using GPs in tandem with glycosyltransferase. In these examples, the GPs were, indeed, used in the phosphorolysis direction, in order to increase the pool of sugar-phosphates from the phosphorolysis of low-cost oligosaccharides (e.g., sucrose in [74]). In these processes, sugar-phosphates are first converted to nucleotide-activated sugars, before being used as substrates for high added-value oligosaccharide/glycoconjugate synthesis by recombinant glycosyltransferases.

When using GPs in the reverse phosphorolysis direction for glycoside synthesis, one issue is the reversibility of the reaction, whcih could lead to low production yields when reaching equilibrium. A typical solution is to shift the equilibrium to promote the reaction in the desired direction, by withdrawing the reverse phosphorolysis product (Figure 4). This can easily be achieved in biphasic systems, when the product presents low solubility in the aqueous medium. Cellodextrin or cellobiose, for instance, produced with the GH94 cellodextrin phosphorylase, show a low solubility in the reaction medium and precipitate, so it can easily be recovered [243,244]. Biomineralization of phosphate under the form of struvite crystals (MgNH_4_PO_4_) has also been reported [245]. Another method for shifting the reaction equilibrium towards substrate consumption is to swap glycosyl phosphates for glycosyl fluorides as donor substrates. The release of fluoride does not cause an attack of the glycoside; thus, preventing phosphorolysis and significantly increasing the production yields [74]. However, glycoside fluorides are more expensive than their phosphate counterpart, and similarly to the fluoride donors for glycosynthases mentioned above, they are often unstable. However, when the desired product remains in the same phase as the substrates, the simplest way to shift the equilibrium towards completion remains to increase one of the substrates’ concentration. Obviously, the cheapest solution is often to increase the acceptor’s concentration, and relative excesses of 1:1.2 to 1:1.5 molar equivalents were shown to give appreciable results [246]. Multi-enzymatic cascades can lead to a similar result, without actually changing the initial concentration of substrates. Due to the interplay of multiple enzymes, the global equilibrium can be substantially changed. For example, in vitro continuous flow phosphorylation of the glycosyl donor using an immobilized phosphorylase [247] allowed successfully synthesizing oligosaccharides by immobilized multiple enzymes cascades [248] or using one-pot multi-enzymatic (OPME) systems [238,249]. Shifting the reaction’s equilibrium towards synthesis can also be successively achieved using an in vitro/in cellulo metabolic coupled system if part of the enzymatic cascade involves cellular catabolic enzymes. This is well exemplified by the synthesis of mannobiose, mannotriose, and mannosyl-glycerate from starch [250]. In this example, six consecutive in vitro steps allow the efficient generation of mannose from starch. This molecule provides the starting material for the intracellular production of β-1,2 mannobiose and mannotriose catalyzed by mannobiose phosphorylase in a *Corynebacterium glutamicum* strain. The synthetic pathway for mannosylglycerate synthesis has also been constructed by channeling the mannose taken up by the strain towards the production of GDP-mannose, the substrate of mannosylglycerate synthase. This can also be fully achieved in cellulo, but this generally requires strain engineering to increase the overall yield or to add the missing steps required to produce certain specialty compounds. Good examples include the production of 2-fucosyllactose, kojibiose or cellobiose using sucrose phosphorylase acting in the phosphorolysis direction in tandem with glycosyltransferases; or even by using two different GPs, one acting in the phosphorolysis direction, to accumulate the desired glycosyl-phosphate, and the other in the synthesis direction [251].

Lastly, when native enzymes do not meet the criteria to be considered for industrial scale production of carbohydrates, they can be modified using enzyme engineering. It is, therefore, possible to broaden the range of application of GPs, by altering their specificity towards defined donors or acceptors, or even improve the enzyme (thermo)stability or resistance to solvents, making them more suited to, or compatible with, certain industrial steps. Reviews illustrating various approaches and examples of GP engineering were recently published [2,32,180]. For example, a lactose phosphorylase was created by mutating the acceptor binding site of a GH94 cellobiose phosphorylase, to extend the range of recognized acceptors [252]. Similarly, the acceptor specificity of a cellobiose phosphorylase was modified in such a way that the engineered enzyme preferred methyl β-glucoside rather than the natural acceptor, glucose [253]. Other examples include *Bifidobacterium adolescentis* sucrose phosphorylase engineering, where a point mutation triggered a domain shift essential to accommodate a variety of large phenolic compounds, permitting resveratrol and quercetin glucosylation [254]. As industrial processes are usually performed near 60 °C, to avoid microbial contamination [255], several studies focused on the discovery and engineering of GPs from thermophilic microorganisms [110]. For instance, the sucrose phosphorylase from *Bifidobacterium adolescentis* showed high temperature tolerance properties compared to the other characterized sucrose GPs, making it a suitable template for thermostability engineering and understanding [32]. The diversity of known GPs is unfortunately restricted to one or a few enzymes for each family, limiting the understanding of their structure–specificity–stability relationships, and, thus, impeding rational engineering studies. Efforts are needed to discover new GP candidates for industrial applications. Even if the very nature of their catalytic mechanism makes them barely distinguishable from canonical GHs and GTs, a handful of approaches allow their detection using activity- and sequence-based methods.

### 2.5. Methods for Glycoside Phosphorylase Activity Detection

Enzyme engineering and discovery from natural sources require the use of suitable screening methodologies. As the number of candidates to be tested can reach hundreds of millions, especially when mining metagenomic or combinatorial mutagenesis libraries, efficient screening methods that allow the detection of enzymes presenting the desired properties must be employed. A variety of approaches, including chromogenic assays, and chromatographic or spectroscopic analyses, are typically employed for detecting GPs and can be adapted to high throughput (Figure 5).

#### 2.5.1. Chromogenic Assays

As the inorganic phosphate released during the reverse-phosphorolysis reaction or consumed during phosphorolysis directly reflects GP activity, measuring its concentration via chromogenic assays constitutes a very convenient proxy to identify GPs. The molybdenum blue activity assay was initially developed by Fiske and Subbarow, but not specifically for detection of GP activity [256]. With this method, the measurement of phosphate consumption or release, depending on the reaction screened for, is based on its conversion to a phosphomolybdate complex in the presence of the molybdate reactant. Under acidic conditions, phosphomolybdate is reduced to molybdenum, of which the blue color intensity can be quantified by absorbance measurement at 655 nm. This method was further modified in several ways, to adapt it to the micro-plate format and automate it [257,258,259]. It was later adapted to the detection of GP-catalyzed reverse-phosphorolysis activity in crude cellular extracts of *E. coli*, which cause high background and consumes glycosyl phosphates [79,260].

Phosphate concentration can also be measured with another chromogenic assay in liquid medium, using coupled enzyme systems. For instance, sucrose phosphorylase activity can be detected using a coupled assay with a phosphoglucomutase (which isomerizes G1P to G6P) and a glucose-6-phosphate dehydrogenase to release 6-phosphogluconate and NADPH. A final step involves a 6-phosphogluconate dehydrogenase yielding an additional NADPH molecule, together with ribulose-5-phosphate [261]. Similarly, the glyceraldehyde-3-phosphate dehydrogenase uses glyceraldehyde-3-P, NAD^+^ and inorganic phosphate to form 1,3-bisphosphoglycerate, releasing H^+^ and NADH. Both NADH and NADPH can then be detected at 340 nm [262]. It is, therefore, possible to couple this conversion to a preceding reaction that releases inorganic phosphate, resulting in an absorbance readout of NADH directly equivalent to the activity of the phosphorylase. Finally, a coupled enzymatic assay usable at pH 6 to 9 was recently developed and established in the context of DNA polymerase. It is based on a three-enzymes system: the concerted action of an inorganic pyrophosphatase, a purine nucleoside phosphorylase, and a xanthine oxidase generate uric acid. Absorbance readout at 293 nm allows monitoring the reaction and directly linking it to the release of phosphate [263]. Both assays are yet to be used to detect GP activity.

More recently, another assay in liquid format was employed for the screening of GP-catalyzed phosphorolysis reactions. It relies on the cleavage of a 2,4-dinitrophenyl glycoside donor in the presence of inorganic phosphate, yielding phenols that could be revealed by absorbance measurement at 400 nm. Even though compatibility with some GP activity detection was proved [30], others can be inactive or weakly active [109] on those chemically modified substrates that differ from the natural glycoside target.

#### 2.5.2. Chromatographic and Spectroscopic Methods for Carbohydrate Detection and Characterization

In order to detect and characterize carbohydrate structures, various methods, with variable sensitivity, are available (Figure 5). These methods drastically vary, regarding e.g., the time for analysis, the sensitivity, the amounts of analyte required, or their destructive or non-destructive nature.

Thin layer chromatography (TLC) is a qualitative separation method that is widely applied for the separation of monosaccharides, oligosaccharides, and sugar phosphates. It is cheap and fast, easy to implement and perform, but suffers from a relative low separation efficiency and lack of information when dealing with unknown spots [264].

High performance liquid chromatography (HPLC) can be used as a quantitative analysis methodology. It generally relies on column-based separation, for example, with copolymer resins in protonated or metal ion forms that are suited for carbohydrate separation. While it offers good separation of carbohydrates (often after optimization of the elution gradient), one significant drawback is carbohydrate detection. Carbohydrates can commonly be detected using low-wavelength UV, but due to the similar absorbance between negatively charged sugars and the organic mobile phase (e.g., acetonitrile), UV detection is unsuited for detecting carbohydrates in mobile phases. Derivatization of the product by adding suitable chemical groups that can be detected at a different wavelengths [265] constitutes a way of dealing with this problem, but this requires tedious additional steps and is, therefore, restricted to qualitative analyses. Another common system for carbohydrate detection relies on the refractive index difference using refractive index detection (RI), although this often suffers from low sensitivity.

High-performance anion-exchange chromatography (HPAEC) offers high sensitivity for the separation of mono-, oligo-, and polysaccharides, based on their structural features. For example, several degrees of polymerization (DP) 2 or DP3 oligosaccharides, only differing by their osidic linkage types, can be separated by HPAEC. This is often coupled with pulsed amperometric detection (PAD), a highly specific and sensitive detection method (in the nanomole range) for carbohydrates [266]. HPAEC-PAD remains one of the most attractive and powerful methods for carbohydrate analysis. Nevertheless, as the response factor depends on the carbohydrate structures, it is impossible to determine the concentration of a product for which no pure standard is available. Although HPAEC allows separating glycosyl phosphates from low DP glycosides and monosaccharides, the very low PAD response factor for glycosyl phosphates makes the simultaneous determination of monosaccharide, oligosaccharide, and glycosyl phosphate concentrations difficult.

The determination of oligosaccharide structures requires information about the anomers, linkages, and their component glycosyl units. Permethylation analysis was developed in the late 1960s and is used to determine the position of glycosidic linkages [267]. Typically, the carbohydrates are derivatized to form acid-stable methyl ethers, then hydrolyzed and converted to alditol acetates, further separated by gas chromatography, and finally analyzed by mass spectrometry (MS) [268].

Nuclear magnetic resonance (NMR) spectroscopy can provide the configuration of glycosidic linkages, by comparing the chemical shifts with those of pure standard molecules. It is non-destructive, highly sensitive, and allows the analysis of product mixtures, even in crude cell extracts [269]. NMR is extensively used to determine the structure of homo-oligosaccharides and homo-polysaccharides synthesized by GPs [101,109,270].

Mass spectrometry (and more specifically, tandem MS) is a powerful analytic method allowing the identification of ionized species according to their mass-to-charge (*m/z*) ratios [271], and which is widely used to determine glycoside structures. The methylated alditol acetates are, therefore, identified by a combination of the relative retention times of the analytes and their fragmentation spectra.

Formal identification of donors and acceptors or osidic linkage type still relies on the use of the traditional workhorses of glycobiology, such as HPAEC-PAD or NMR. Reverse-phosphorolysis product structural determination also relies on a standard’s commercial availability or chemical synthesis. When no standards are available, highly resolutive MS-MS can be used to determine the linkage position. Recent methods, such as ion-mobility based separation, offer exquisite carbohydrate separation, even for isomers co-eluted by other separation techniques [272]. Associated with very sensitive and specific tandem MS methods, (complex) carbohydrate structure can now be completely elucidated [272,273]. This analysis only requires 1 µg of carbohydrate, directly from reactional mixtures. This new analytical technique paves the way for carbohydrate ‘sequencing’ using cyclic ion mobility mass spectrometry. Overall, great improvements in these methodologies allow a faster separation and the handling of more samples (with data acquisition per sample within minutes), with reduced compound consumption. These methods, previously used for precise product characterization in final screening steps, now offer real possibilities for screening large libraries, while generating good quality data [113].

### 2.6. Enzyme Discovery by Functional (Meta)Genomics

The vast majority (74%) of GPs discovered so far originate from archaebacteria and bacteria, which produce these enzymes for glycoside catabolism. More than 10^30^ bacterial and archaeal cells are present on Earth [274]. Depending on the habitats, uncultured species can make up to 99.9% of the ecosystem [270]. For instance, the human gut microbiota, one of the most studied ecosystems, due to its importance for health, counts about 70% uncultured species [275]. With the rapid development of next generation sequencing (NGS) technologies, sequencing costs have significantly dropped in the past two decades, opening the way to massive sequencing of metagenomes, the collective genomes of microbiota [276]. Most of these works are monogenic metagenomic studies, focusing on the sequencing of 16S (bacteria, archea) and 18S (eukaryotes) rRNA genes, to describe the microorganisms diversity in an environmental sample, without culturing them [277]. Nevertheless, the last decade has seen more and more functional metagenomics projects being launched. Functional metagenomics is the sequencing of functional genes to predict and/or prove their function [278]. These projects led to the establishment of several gene catalogs, issued from various microbiota, particularly those of human [279,280], mouse [281,282], pig [283], and cow [284] guts; totaling nearly 34 million genes and thousands metagenome-assembled genomes (MAGs). The drastic increase of available metagenomic sequences from various catalogs yielded hundreds of novel enzymes and activities, identified through activity- or sequence-based analyses (Figure 5).

#### 2.6.1. Activity-Based Approaches

The activity-based approaches’ typical workflow begins with metagenomic DNA cloning (or cDNA for eukaryotes) from metatranscriptomic samples, in a vector (plasmids, fosmids, or bacterial artificial chromosomes). The genes issued from the microbiome are then expressed in a recombinant host [285]. In activity-based metagenomic studies, the host is often *Escherichia coli*, due to its high transformation efficiency and its ability to efficiently express genes issued from distant taxa [286]. The metagenomic libraries then have to be screened with a suitable screening assay, preferably at a high throughput. Activity-based approaches raise issues, such as the typical GC bias found in *E. coli* clone libraries, which may affect the DNA insert stability in the vector [287], or the potential toxic effect of the expressed exogenous gene on the cells.

To circumvent these problems, other host strains and vectors have been developed [288,289,290]. In activity-based metagenomics, the positive hit rate greatly depends on the metagenomic DNA source, its abundance in the metagenomic sample, the chosen cloning vector and expression host, the variability of heterologous gene expression, and the sensitivity and inherent bias of the screening assay. Hit rates typically range from 0.01‰ to 10‰ [291]. This approach, thus, requires high-throughput screening facilities to screen a sufficiently large sequence space (typically 10^4^ − 10^5^ clones, which corresponds to 0.3 × 10^6^ to 3 × 10^6^ genes for fosmid libraries). Depending on the assay format, the screening throughput varies between 10^4^ (automated liquid assays in micro-plates) to 10^5^ assays per day (automated assays on solid plates) [292], up to 10^6^ with droplet-microfluidics [293].

To date, only three activity-based functional metagenomic studies have been performed to discover GPs. In the first one, a new sucrose-phosphorylase (Genbank accession FJ472846, not listed in the CAZy database, but likely a GH13_18 member) was isolated from a 5000-clone metagenomic library constructed from a sucrose refinery sample. In this case the screen consisted in a positive selection assay on a solid mineral medium containing sucrose as sole-carbon source [49]. More recently, the S. Withers group has started to use metagenomics to mine various ecosystems for GPs. Their first study was based on the screening of 17,168 clones library obtained from a passive mine tailings biochemical reactor system, on the activated substrate 2,4-dinitrophenyl β-d-glucoside (DNPGlc) and inorganic phosphate, leading to the discovery of the retaining β-glucoside-phosphorylase BglP, a rare GP from the GH3 family [30]. The last study focused on cello-oligosaccharide-degrading GPs, searched in ∼23,000 clones sourced from the same passive mine tailings biochemical reactor system, as in the previous study, and from beaver feces. In this work, MacDonald and colleagues used a liquid assay based on molybdenum blue formation to identify seven new GPs from the GH94 and GH149 families (cellobiose phosphorylases, cellodextrin phosphorylases, laminaribiose phosphorylases, and a β-1,3-glucan phosphorylase) [79]. By varying substrate combinations, the authors demonstrated the versatility of the screening method, with which we can expect GPs from other families to be discovered in the near future.

#### 2.6.2. Sequence-Based Approaches

Contrary to activity-based approaches, sequence-based screening is performed using either in silico sequence homology-based methods, or in vitro, using DNA/DNA or DNA/RNA hybridization-based methods (such as PCR) to isolate sequences with known consensus motifs. The latter have only been used to discover CAZymes [294], and not yet to discover GPs. Since they rely on identifying sequences that possess the same motifs involved in substrate recognition and catalysis as those of already known enzymes, these methods perform poorly in discovering new enzyme activities and families.

In silico sequence homology-based methods rely on isolating sequences similar to a reference sequence for which the function has been experimentally confirmed, or which differs from it at particular locations. Two highly similar sequences are indeed expected to have evolved from a common ancestor, and, thus, most likely share a similar function. Alignment-search tools such as BLAST [295] are used to classify proteins into evolutionary families. It is, however, possible that two distant protein sequences may have the same function, owing to structurally conserved active site motifs. This is actually the case for members of the different CAZy families that share the same activity. For instance, β-mannosidases can be found in five different families (GH1, GH2, GH5, GH130, and GH164), even though some are structurally unrelated; with the GH1, GH2, and GH5 families harboring (β/α)_8_ folds and GH130 the five-bladed beta-propeller fold (unknown for GH164). In some cases, the opposite is found: the CAZy families may be related distantly, sharing less than 20–30% sequence identity, but still have high structural similarities, as for GH130 and GT108 families, which both have a five-bladed beta-propeller fold, and which both contain β-mannoside-phosphorylases [174].

Several databases, including COGs, Pfam, and KEGG, allow for automated functional annotation of metagenomic sequences. COGs identifies orthologous genes that typically share the same general function, such as carbohydrate transport and metabolism for the G cluster, in which CAZymes are found [296]. Pfam contains nearly 18,000 protein families (Pfam 32.0) [297]. Each family is created based on multiple sequence alignments and hidden Markov models (HMMs) [298]. Since Pfam families (like CAZy families) contain members with several activities, the activity of an enzyme cannot be deduced from the family it belongs to. Finally, KEGG can provide information on the involvement of a protein in the metabolism of an organism or an ecosystem [298]. Nevertheless, sequence similarity-based methods leave orphan genes (genes which do not share significant sequence similarity with any known sequence) without annotation. Furthermore, due (and participating in) to the amplification of incorrect functional annotations in databases, these tools can generate annotation mistakes in the preliminary functional annotation of sequences. For instance, when searching for the GH130 β-1,4-mannoside-phosphorylase Uhgb_MP (Genbank accession number ADD61463.1) in these databases, the results are the following:COG: the protein belongs to COG2152: Predicted glycosyl hydrolase, GH43/DUF377 family (Carbohydrate transport and metabolism). The biochemical characterization of this enzyme [109], indicated that these results are false, this enzyme being a mannoside-phosphorylase of the GH130 family, which does not contain a GH43 module.Pfam: the protein contains the domain Glyco_hydro_130 (PF04041), described as ‘beta-1,4-mannooligosaccharide phosphorylase’. This is true, but the same result is found for the GH130 β-1,2-Mannosidase AAO78885.1 [194].Kegg: no result.

The IMG-M and MG-RAST servers [299,300] and the commercially available QIAGEN CLC Microbial Genomics Module are convenient tools which integrate data resulting from the interrogation of these different databases. They make it possible to quantify and compare the abundance of the main functional families in the target ecosystems.

For more accurate function prediction, especially for substrate specificity or mechanism prediction of an enzyme, sequence analysis can be complemented with local genomic neighborhood analysis. Indeed, the in silico analysis of polysaccharide utilization loci (PUL) composition, and thus, of the combinations of CAZymes involved in glycoside catabolism, can be a good way to predict the substrate targeted by a Bacteroidota PUL [301], and by each of its CAZyme components. This helped to discover the biological function of Uhgb_MP (for ‘unknown human gut bacterium_mannoside-phosphorylase’), the first GP issued from an uncultured bacterium [109], thanks to the cloning of entire metagenomic PULs in fosmids [286]. More recently, this strategy permitted elucidating the function of a novel sucrose-6^F^-phosphate-phosphorylase from the cultured human gut bacterium *Ruminococcus gnavus* E1 [44]. Nevertheless, such a strategy is restricted to long DNA sequences, and cannot be used with highly fragmented loci sequences, such as those found in datasets obtained by massive and random sequencing of metagenomes. Moreover, when multiple uncharacterized GPs from the same family are found together on the same PUL, predicting the substrate nature is very difficult, making the results uncomplete. Besides, given that the Bacteroidota PULs are the sole CAZy-containing bacterial operons referenced to date, the prediction of non-Bacteroidota PUL-like systems remains highly speculative, resulting in a large variability of the loci composition.

Mining unexplored clades of GP-containing CAZy families or subfamilies phylogenetic trees is also an efficient strategy for discovering new GP specificities, as exemplified with the GH13_18 subfamily [180]. This strategy led to the discovery of glucosylglycerate phosphorylases [28,42] and of a sucrose 6′-phosphate phosphorylases [48]. In the GH130 family, β-1,3- [102] and β-1,2-mannoside-phosphorylases [101] were found in the genomes of cultured strains.

Structure prediction and docking experiments can also be very useful [302]. Examination of the active site and catalytic residues conservation is indeed probably the best way to predict if a CAZyme is a GH or a GP. Inverting GH130 enzymes can either be GHs when they possess the two canonical carboxylic amino acids required to catalyze hydrolysis reactions, or GPs when only the proton donor and phosphate-binding residues are present [109,194]. For retaining enzymes, some GH3 members were shown to act as phosphorylases rather than hydrolases, because they use a His/Asp dyad as an acid/base catalyst rather than the standard Glu or Asp residues. This allows an anionic phosphate to enter the active site; while this is greatly disfavored when the active site bears a deprotonated carboxylate base directly adjacent to the anomeric center of the glycosyl enzyme [31]. This difference was confirmed recently, when a single point mutation of the acid/base, from Asp to Asn was shown to convert GH84 retaining hydrolases to efficient phosphorylases [176]. However, in-depth sequence analysis and structure prediction require specific expertise, to handle large amounts of data and align thousands of sequences.

Analyzing such high numbers of sequences more easily than by using multiple sequence alignments and phyologenetic tree analyses, to visualize sequence relationships in protein families, is essential. The Enzyme Function Initiative-Enzyme Similarity Tool (EFI-EST) was developed in 2015 [303] for this purpose. This web tool, dedicated to all members of the biological/biomedical community, including non-bioinformaticians, automatically generates sequence similarity networks (SSNs) [304], based on pairwise sequence comparison. In an SSN, each sequence is represented by a node (symbol), connected by an edge (line) to all the other nodes corresponding to sequences that share a sequence similarity greater than a user-specified value. This approach allows highlighting unexplored sequence spaces, guiding biochemists to discover novel functions. SSNs have been used to split a glycoyl radical enzyme superfamily into different iso-functional clusters, guiding the biochemical characterization efforts towards unexplored clusters of metagenomic sequences that are highly abundant in the human gut microbiome. This led to the discovery of a novel enzyme involved in the biosynthesis of L-proline, a key mediator of healthy microbiota–host symbioses [305]. Even though extremely powerful, the SSNs-based approach had not been tested for CAZyme diversity analysis until very recently, with the multispecific GH16 family subdivision into 23 robust subfamilies [306]. The GH130 family was also recently analyzed with this tool, which allowed revisiting the historical GH130_1 and GH130_2 subfamilies classification, to generate 15 robust sequence clusters (20+ sequences containing meta-nodes) [112]. This structure–sequence–function data integration permitted focusing biochemical characterization efforts onto five cluster representatives, to demonstrate the segregation of hydrolases and phosphorylases at the cluster level, and identify new functions, such as the GH130 Man-GlcA phosphorylase [112]. Finally, we can report the discovery of a new GP activity among GH94 enzymes using an SSN based strategy, 4-O-β-d-glucosyl-d-galactose phosphorylase [86]. However, the CAZy database and the publicly available metagenomic sequence datasets remain largely untapped, representing gold mines to be explored with such approaches, with the prospect of novel GP discoveries.

## 3. Conclusions and Perspectives

The necessary conversion towards eco-friendly solutions and cleaner processes in industry is largely based on the use of biocatalysts. Renewable sourcing of carbon mostly relies on the use of plant polysaccharides, meaning that an increasing number of processes will be dependent on CAZymes. Glycoside phosphorylases are fascinating CAZymes, with unique properties, combining the best of both worlds of GHs and GTs. The integration of such enzymes into the expanding catalogue of industrial applications, envisioned through the march towards global decarbonization, is a challenge facilitated by the exquisite plasticity and reactional diversity provided by these energy-optimized catalysts. Their unique biochemical properties, making them particularly adapted for biochemical processes improvements, have long been recognized, as testified by the interest in these enzymes manifested by both the scientific community and private companies. They, indeed, provide an elegant way to address the shortcomings of GHs and GTs, either by a direct replacement at the reaction level for improved efficiency or by complementation in fully integrated biological pathways.

Mostly used for their synthetic properties, polysaccharide degradation by GPs is energetically superior to hydrolysis, since transferring a glycosyl moiety onto a phosphate residue, rather than onto a water molecule, retains the inter-osidic bond energy in the newly created glycosyl-phosphate bond. The generated glycosyl-phosphate can subsequently be reused as an activated donor, by the same or another GP. Initially produced from a generally cheap and renewable material, this is a sustainable and elegant way to generate activated sugars. These are, most of the time, directly used as substrates for the actual reaction of interest, with the transfer of the glycosyl moiety onto an exogenous acceptor. Molecules of interest are extremely diverse, whether for the production of bio-compatible compounds designated for food/health applications or as an alternative for hazardous substances. GPs can also be part of a larger in vitro synthesis chain of reactions, helping to reduce the use of solvents, or improving the overall yield by providing exquisite stereoselectivity at a critical step. Nevertheless, even when only used for the degradation of polysaccharides, the energetic superiority of GPs over GHs can make a difference if the resulting monomers are to be fermented for direct energy supply, or as input carbohydrates for in cellulo synthesis in microbial cell factories, since sugar-phosphates have a higher energetic content than their isolated monosaccharide equivalents.

Nevertheless, the full potential of GPs remains largely underexploited, due to the very nature of their similarity to both GHs and GTs. As sequence-specific characteristics that would make them easy to identify are limited, case-by-case manual testing largely remains the method employed to demonstrate GP activity and characterize their acceptor tolerance. Thus, the known GP-catalyzed reactions are still largely less diverse than those catalyzed by GHs, and studies aiming at specifically isolating them from natural environments are also less common. However, in recent years technological advances have been made to address these limitations. GP-specific biochemical assays have been developed and used in high-throughput screening campaigns. Droplet microfluidics, in this respect, holds great promise for GP discoveries by functional metagenomics. Sequence-specific markers, such as the absence of signal peptides or unique phosphate binding residues can so far only be analyzed manually, family by family, when functional data are available. Multi-omics technologies can now be augmented by automated genomic data analysis, using methods such as SSNs, allowing glimpsing a bright future in the field, by specifically identifying potential GP-containing sequence clusters. These tools provide guidance to the experimenter, for focusing their efforts onto the most promising targets identified in the mountains of genomic data accumulated over the years. Reanalyzing GP-containing CAZy families in this new light would also help identifying novel activities from isolated sequence groups. SSNs are not only advantageous in highlighting groups of uncharacterized sequences, but also by splitting the diversity of each family into several smaller sequence groups, facilitating multiple sequence alignments, and searching for consensus motifs indicative of, e.g., GPs versus GHs or a particular activity. Splitting large datasets into sub-groups of similar sequences with SSNs is also an interesting strategy to decrease computational time and to improve structural alignments, based either on crystallographic protein structures or on tridimensional models. Indeed, SSNs alone cannot be used to predict enzyme mechanism, nor substrate or linkage specificities, which can vary due to differences in a single amino acid. When a sufficient number of enzymes with different mechanisms and linkage specificities have been biochemically characterized, primary sequence analysis can be sufficient to predict the GP or GH mechanism.

However, most of the time, SSNs and multiple sequence alignment analysis are insufficient to predict substrate specificities, since discrete differences of the active site topologies between enzymes belonging to the same meta-node can result in different activities. SSNs also have other limitations. First, they cannot be directly used to analyze CAZy families containing many multi-modular sequences. In this case, the multi-modularity must be analyzed and the sequences truncated, to analyze with SSNs, only the CAZy module of interest. This is what was done with the GH16 family [306]. Nevertheless, as a large share of GPs families are mono-modular, SSNs can often be used directly. The other limit of SSNs is their instability when the number and diversity of the analyzed sequences vary. This is why the creation of stable CAZy sub-families cannot be based solely on SSNs-based clustering, but must also use phylogenetic analyses, which are much more stable and have the advantage of providing evolutionary distances, contrary to SSNs. Nevertheless, again, SSNs can be used to facilitate the creation of the multiple sequence alignments required for the generation of such phylogenetic trees, by selecting a smaller number of sequences from each meta-node.

Artificial intelligence (AI)-based tools have attracted a lot of attention in recent years, holding promise for faster and more accurate structural predictions. Equipped with improved structural modeling tools, structural biology is thought to ease GP identification directly from protein sequences. Highlighting structural hotspots for GP activity or substrate specificity is thought to convey meaningful information for achieving fast and reliable production of desirable catalysts. AI is also envisioned to be an extremely valuable tool for data mining and integration, making sense of dispersed data by facilitating the combination of multiple data types and the summarization into structural projections easily understandable by the experimenter.

Ultra-high throughput (UHT) functional screening technologies, such as droplet microfluidics, which have only started to be used for CAZyme screening, have vast potential for GP identification and engineering [307]. Using, e.g., the coupled reactions detailed previously in this manuscript, it might be possible to specifically target phosphorolytic activities from libraries > 10^6^ members large, making full use of the increased throughput offered by droplet microfluidics. With in-line coupling with analytical methods, such as MS or NMR, droplet microfluidics can become a valuable tool for glycomics, allowing the sorting of library members directly upon structural identification of the desired glycan patterns produced.

## Figures and Tables

**Figure 1 ijms-23-03043-f001:**
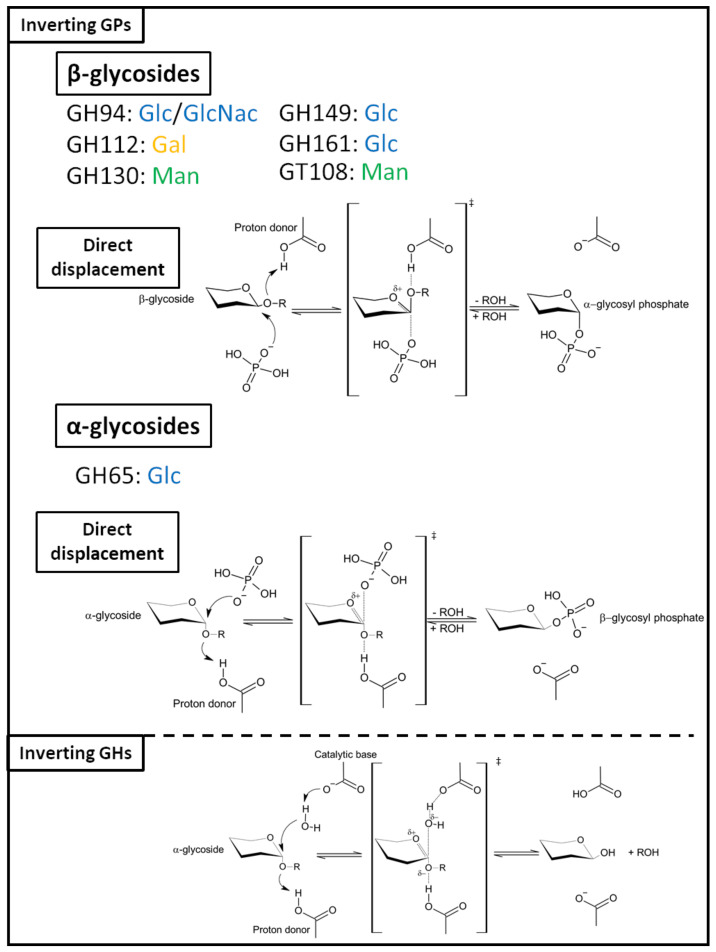
Inverting GPs and GHs catalytic mechanisms and family classification. The carbohydrates indicated for each family correspond to the glycosyl donors used by GPs of this family.

**Figure 2 ijms-23-03043-f002:**
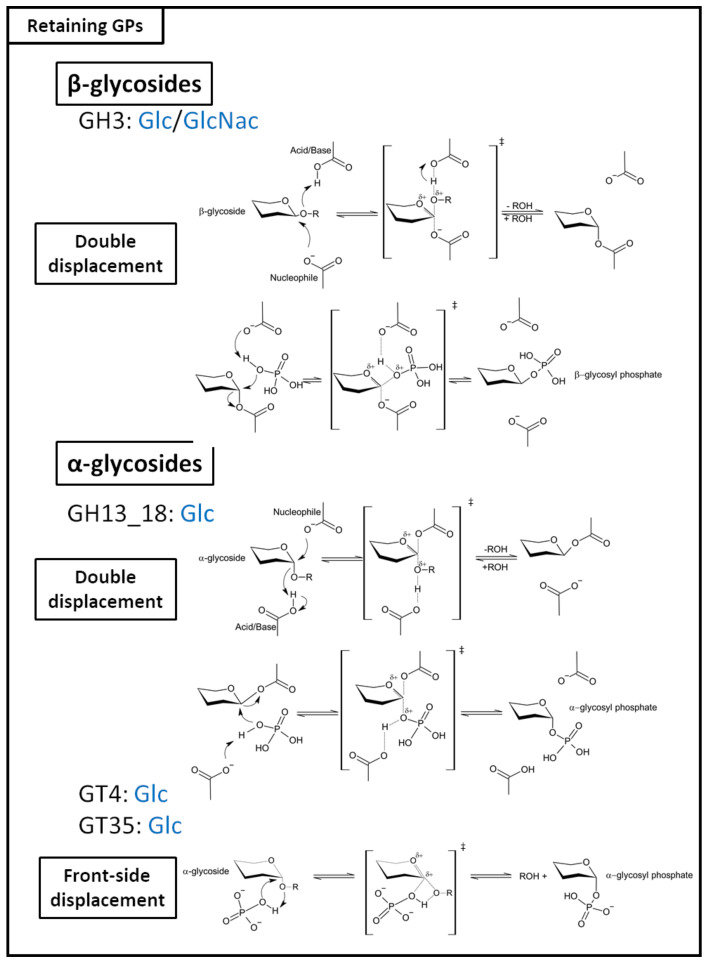
Retaining GPs catalytic mechanisms and family classification. The carbohydrates indicated for each family correspond to the glycosyl donors used by GPs of this family.

**Figure 3 ijms-23-03043-f003:**
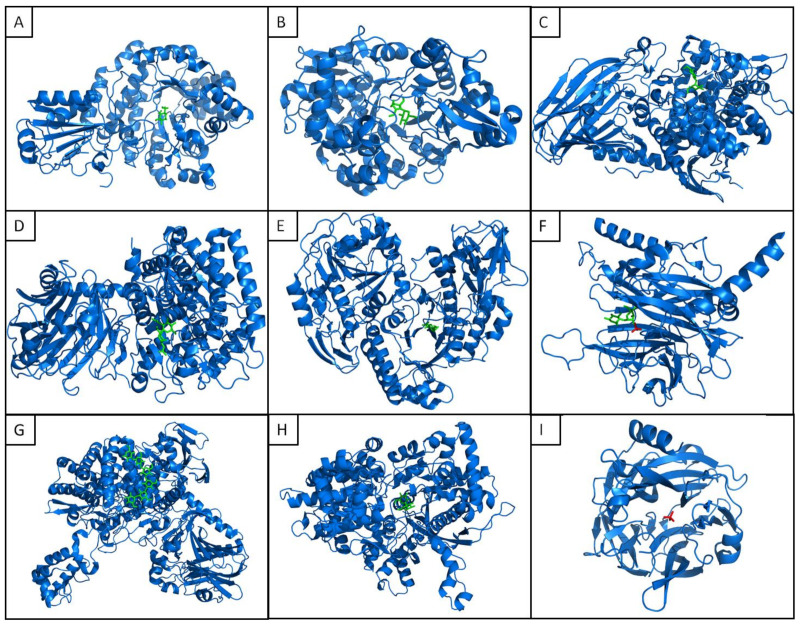
Structural representatives of the phosphorylase families with their main substrate (glycoside donnor, or acceptor, in green, or inorganic phosphate in red, when available). (**A**) GH3 β-glucoside phosphorylase BglX bound to 2FGlc (5VQE); (**B**) GH13_18 sucrose phosphorylase E232Q mutant from *Bifidobacterium adolescentis* in complex with sucrose (2GDU); (**C**) GH65 kojibiose phosphorylase from *Caldicellulosiruptor saccharolyticus* DSM 8903 in complex with kojibiose (3WIQ); (**D**) GH94 chitobiose phosphorylase from *Vibrio proteolyticus* in complex with GlcNAc (1V7W); (**E**) GH112 galacto-*N*-biose phosphorylase from *Bifidobacterium longum* subsp. longum JCM 1217 in complex with GalNAc (2ZUT). (**F**) GH130 MannosylGlucose phosphorylase from *Bacteroides fragilis* NCTC 9343 in complex with mannosyl-glucose and phosphate (3WAS); (**G**) GH149 β-1,3-oligoglucan phosphorylase in complex with laminaribiose (6HQ8); (**H**) GT35 maltodextrin phosphorylase from Escherichia coli str. K-12 substr. MG1655 in complex with maltose (1AHP); (**I**) GT108 Dual-Activity Glycosyltransferase-Phosphorylase from *Leishmania mexicana* MHOM/GT/2001/U1103 in complex with inorganic phosphate (6Q50).

**Figure 4 ijms-23-03043-f004:**
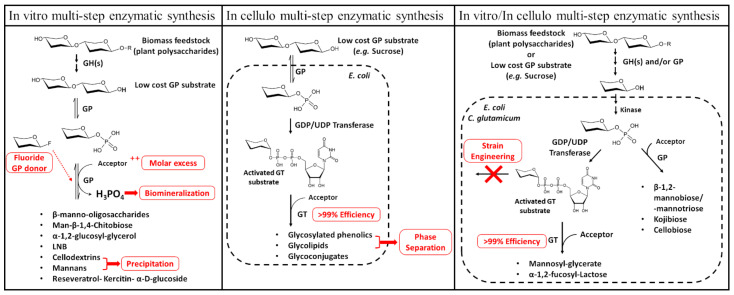
Biotechnological applications of GPs: the different strategies employed for improving glycoside production yields are marked in red.

**Figure 5 ijms-23-03043-f005:**
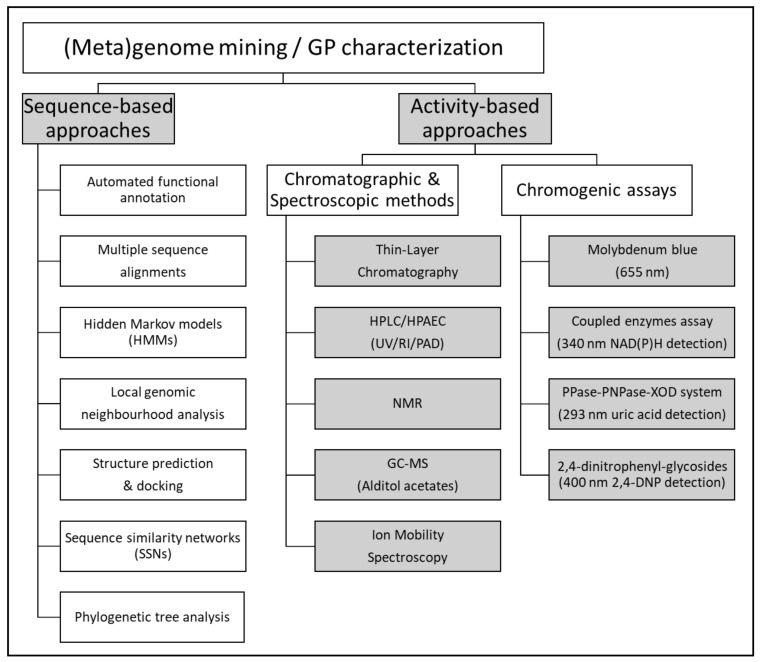
Sequence- and activity-based approaches for GP discovery from (meta)genomes and functional characterization.

**Table 1 ijms-23-03043-t001:** List of functionally characterized GPs. Some of the enzymes (labelled with a star) referenced in this table do not currently appear in the list of characterized enzymes of the CAZy Database.

Family	Mechanism	GenBank Uniprot	Protein Name	Organism	PDB	Reverse-Phosphorolysis Activity			Reference (DOI)
						Donor	Acceptor	Product	
GH3	Retaining	AUG44408.1	β-glucoside phosphorylase (BglP)	uncultured bacterium	5VQD, 5VQE	β-GlcNac1P β-Glc1P	*p*-NP/DNP	*p*-Nitrophenyl-*N*-acetyl-β-d-glucosaminide *p*-nitrophenyl-β-d-glucopyranoside 2,4-Dinitrophenyl-β- d-glucopyranoside	[30]
GH3	Retaining	AAQ05801.1	Bifunctional *N*-acetyl-β-glucosaminidase/β-glucosidase (Nag3)	*Cellulomonas fimi* ATCC 484		β-Glc1P	DNP	2,4-Dinitrophenyl-β-d-glucopyranoside	[31]
GH13_18	Retaining	AAO33821.1	Sucrose phosphorylase (SucP;SP;BaSP)	*Bifidobacterium adolescentis* DSM 20083	1R7A, 2GDU, 2GDV, 5C8B, 5M9X, 5MAN, 5MB2, 6FME	α-Glc1P	d-fructose	Sucrose: (α-d-glucopyranosyl-(1→2)-β-d-fructofuranoside)	[32]
GH13_18	Retaining	AAN24362.1	Sucrose phosphorylase (Spl;BL0536)	*Bifidobacterium longum* NCC2705		α-Glc1P	d-fructose	Sucrose	[33]
GH13_18	Retaining	AAO84039.1	Sucrose phosphorylase (SplP)	*Bifidobacterium longum* SJ32		α-Glc1P	d fructose	Sucrose	[34]
GH13_18	Retaining	BAF62433.1	Sucrose phosphorylase (Spl)	*Bifidobacterium longum subsp. longum* JCM 1217		α-Glc1P	d-fructose	Sucrose	[35]
GH13_18	Retaining	AAC74391.2	Glucosylglycerate phosphorylase (YcjM)	*Escherichia coli str. K-12 substr*. MG1655		α-Glc1P	d glycerate	Glucosyl-glycerate	[28]
GH13_18	Retaining	AAO21868.1	Sucrose phosphorylase (LaSP;GtfA2;LBA1437)	*Lactobacillus acidophilus* NCFM		α-Glc1P	d fructose	Sucrose	[36]
GH13_18	Retaining	BAA14344.1	Sucrose phosphorylase (LmSPase)	*Leuconostoc mesenteroides* ATCC 12291		α-Glc1P	d-fructose	Sucrose	[37]
GH13_18	Retaining	ABS59292.1	Sucrose phosphorylase (742sp)	*Leuconostoc mesenteroides* B-742		α-Glc1P	d-fructose	Sucrose	[38]
GH13_18	Retaining	2207198A	Sucrose phosphorylase	*Leuconostoc mesenteroides* NO. 165		α-Glc1P	d-fructose	Sucrose	[39]
GH13_18	Retaining	AAX33736.1	Sucrose phosphorylase (LmSP1)	*Leuconostoc mesenteroides* NRRL B-1149		α-Glc1P	d-fructose	Sucrose	[40]
GH13_18	Retaining	AGK37834.1	Sucrose phosphorylase (ScrP)	*Limosilactobacillus reuteri* LTH5448		α-Glc1P	d-fructose	Sucrose	[41]
GH13_18	Retaining	ADP98617.1	Glucosylglycerol phosphorylase (HP15_2853)	*Marinobacter adhaerens* HP15		α-Glc1P	d-glycerol	Glucosyl-glycerol: (α-d-glucopyranosyl-(1→2)-glycerol)	[42]
GH13_18	Retaining	ADH62582.1	Glucosylglycerate phosphorylase (MSGGaP;Mesil_0665)	*Meiothermus silvanus* DSM 9946		α-Glc1P	d-glycerate	Glucosyl-glycerate: (α-d-glucopyranosyl-(1→2)-glycerate)	[28]
GH13_18	Retaining	AAD40317.1	Sucrose phosphorylase	*Pelomonas saccharophila* IAM 14368		α-Glc1P	d-fructose	Sucrose	[43]
GH13_18	Retaining	CCA61958.1	Sucrose 6(F)-phosphate phosphorylase (RUGNEv3_61221;SucP)	*Ruminococcus gnavus* E1		α-Glc1P	d-fructose-6-P	Sucrose-6-phosphate	[44]
GH13_18	Retaining	AEJ61152.1	Glucosylglycerate phosphorylase (Spith_0877)	*Spirochaeta thermophila* DSM 6578		α-Glc1P	d-glycerate	Glucosyl-glycerate	[28]
GH13_18	Retaining	CAA30846.1	Sucrose phosphorylase (GftA;SmSP)	*Streptococcus mutans* INGBRITT/GS5		α-Glc1P	d-fructose	Sucrose	[45]
GH13_18	Retaining	AAN58596.1	Sucrose phosphorylase (GtfA;SMU.881)	*Streptococcus mutans* UA159		α-Glc1P	d-fructose	Sucrose	[46]
GH13_18	Retaining	CAA80424.1*	Sucrose phosphorylase	*Agrobacterium vitis*		α-Glc1P	d-fructose	Sucrose	[47]
GH13_18	Retaining	ADL69407.1	Sucrose 6(F)-phosphate phosphorylase (SPP;TtSPP;Tthe_1921)	*Thermoanaerobacterium thermosaccharolyticum* DSM 571	6S9V	α-Glc1P	d-fructose-6-P	Sucrose-6-phosphate	[48]
GH13_18	Retaining	ACJ83248.1	Sucrose phosphorylase (Unspase)	Uncultured bacterium		α-Glc1P	d-fructose	Sucrose	[49]
GH13_18	Retaining	BAN03569.1	Sucrose 6(F)-phosphate phosphorylase (YM304_32550)	*Ilumatobacter coccineus* YM16-304	6S9U	α-Glc1P	d-fructose-6-P	Sucrose-6-phosphate	[50]
GH65	Inverting	AAV43670.1	Maltose phosphorylase (MalP;LBA1870)	*Lactobacillus acidophilus* NCFM		β-Glc1P	d-glucose	Maltose: (α-d-glucosyl-(1→4)-d-glucose)	[51]
GH65	Inverting	ADH99560.1	Maltose phosphorylase (Bsel_2056)	*Bacillus selenitireducens* MLS10		β-Glc1P	d-glucose	Maltose	[52]
GH65	Inverting	BAC54904.1	Maltose phosphorylase (MPase)	*Bacillus sp*. RK-1		β-Glc1P	d-glucose	Maltose	[53]
GH65	Inverting	BAD97810.1	Maltose phosphorylase (MapA)	*Paenibacillus sp.* SH-55		β-Glc1P	d-glucose	Maltose	[27]
GH65	Inverting	CAA11905.1	Maltose phosphorylase	*Fructilactobacillus sanfranciscensis* DSM 20451(T)		β-Glc1P	d-glucose	Maltose	[54]
GH65	Inverting	AAO80764.1	Maltose phosphorylase (MalP;EF0957	*Enterococcus faecalis* V583		β-Glc1P	d-glucose	Maltose	[55]
GH65	Inverting	Q7SIE1	Maltose phosphorylase	*Levilactobacillus brevis* ATCC 8287	1H54	β-Glc1P	d-glucose	Maltose	[56]
GH65	Inverting	BAQ19546.1	Maltose phosphorylase (MalE)	*Bacillus sp*. AHU2001		β-Glc1P	d-glucose	Maltose	[57]
GH65	Inverting	BAB97299.1	Trehalose phosphorylase (TreP)	*Thermoanaerobacter brockii* ATCC 35047	Preliminary data PMID: 20383018	β-Glc1P	d-glucose	Trehalose: (α-d-glucosyl-(1→1)-d-glucose)	[58]
GH65	Inverting	BAC20640.1	Trehalose phosphorylase (TPase)	*Geobacillus stearothermophilus* SK-1		β-Glc1P	d-glucose	Trehalose	[59]
GH65	Inverting	KKC30106.1	Trehalose phosphorylase (TP)	*Caldanaerobacter subterraneus subsp. Pacificus* DSM 12653		β-Glc1P	d-glucose	Trehalose	[60]
GH65	Inverting	ADG89586.1	Trehalose phosphorylase (TbGP; Tbis_2887)	*Thermobispora bispora* DSM 43833		β-Glc1P	d-glucose	Trehalose	[61]
GH65	Inverting	AAK04526.1	Trehalose-6-phosphate phosphorylase (TrePP;YeeA;L39593; LL0428)	*Lactococcus lactis subsp. lactis* Il1403		β-Glc1P	d-glucose-6-phosphate	Trehalose-6-phosphate: (α-d-glucosyl-(1→1)-d-glucopyranoside-6-phosphate)	[62]
GH65	Inverting	APF18594.1	Trehalose-6-phosphate phosphorylase (CaGP;Cabys_1845)	*Caldithrix abyssi* DSM 13497 LF13		β-Glc1P	d-glucose-6-phosphate	Trehalose-6-phosphate	[61]
GH65	Inverting	AAE30762.1	Kojibiose phosphorylase (KojP;KPase)	*Thermoanaerobacter brockii* ATCC 35047		β-Glc1P	d-glucose	Kojibiose: (α-d-glucosyl-(1→2)-d-glucose)	[58]
GH65	Inverting	AAC74398.1	Kojibiose phosphorylase (YcjT)	*Escherichia coli str. K-12 substr*. MG1655		β-Glc1P	d-glucose	Kojibiose	[63]
GH65	Inverting	ABP66077.1	Kojibiose phosphorylase (CsKP;Csac_0439)	*Caldicellulosiruptor saccharolyticus* DSM 8903	3WIQ, 3WIR	β-Glc1P	d-glucose	Kojibiose	[64]
GH65	Inverting	ACL68803.1	Kojibiose phosphorylase (HoGP; Hore_00420)	*Halothermothrix orenii* H 168		β-Glc1P	d-glucose	Kojibiose	[61]
GH65	Inverting	ABX42243.1	Nigerose phosphorylase (Cphy_1874)	*Lachnoclostridium phytofermentans* ISDg		β-Glc1P	d-glucose	Nigerose: (α-d-glucosyl-(1→3)-d-glucose)	[65]
GH65	Inverting	ABX43667.1	Nigerose phosphorylase (Cphy_3313)	*Lachnoclostridium phytofermentans* ISDg		β-Glc1P	d-glucose	Nigerose	[66]
GH65	Inverting	ADQ05832.1	1,3-α-oligoglucan phosphorylase (ChGP;Calhy_0070)	*Caldicellulosiruptor hydrothermalis* 108		β-Glc1P	Maltose Kojibiose Nigerose	α-1,3-oligoglucan	[61]
GH65	Inverting	ADC90669.1	Trisaccharide α-glucan phosphorylase (MiGP;HMPREF0868_0806)	*Mageeibacillus indolicus* UPII9-5		β-Glc1P	Isomaltose Isomaltulose Maltose Kojibiose Nigerose	α-1,3-oligoglucan	[61]
GH65	Inverting	ABX43668.1	α-1,3-oligoglucan phosphorylase (Cphy_3314)	*Lachnoclostridium phytofermentans* ISDg		β-Glc1P	d-glucose	α-1,3-oligoglucan	[66]
GH65	Inverting	ABX41399.1	3-O-α-glucopyranosyl-L-rhamnose phosphorylase (Cphy_1019)	*Lachnoclostridium phytofermentans* ISDg		β-Glc1P	l-rhamnose	α-d-glucopyranosyl-(1,3)-l-rhamnose	[25]
GH65	Inverting	ADI00307.1	α-1,2-glucosylglycerol phosphorylase (Bsel_2816)	*Bacillus selenitireducens* MLS10	4KTP, 4KTR	β-Glc1P	d-glycerol	(α-d-glucosyl-(1→2)-glycerol)	[67]
GH94	Inverting	AAB95491.2	Cellobiose phosphorylase (CbpA)	*Thermotoga neapolitana*		α-Glc1P	d-glucose	Cellobiose: (β-d-glucopyranosyl-(1,4)-d-glucopyranose)	[68]
GH94	Inverting	AAC45510.1	Cellobiose phosphorylase (CepA)	*Thermoclostridium stercorarium* NCIB 11754		α-Glc1P	d-glucose	Cellobiose	[69]
GH94	Inverting	AAD36910.1	Cellobiose phosphorylase (CepA;TM1848;Tmari_1863	*Thermotoga maritima* MSB8		α-Glc1P	d-glucose	Cellobiose	[70]
GH94	Inverting	AAL67138.1	Cellobiose phosphorylase (Cbp)	*Acetivibrio thermocellus* YM4	3QDE	α-Glc1P	d-glucose	Cellobiose	[71]
GH94	Inverting	AAQ20920.1	Cellobiose phosphorylase (CbP)	*Cellulomonas uda* DSM 20108	3RRS, 3RSY, 3S4A, 3S4B, 3S4C, 3S4D	α-Glc1P	d-glucose	Cellobiose	[72]
GH94	Inverting	ABD80580.1	Cellobiose phosphorylase 94A (Cbp;SdCBP;Sde1318, Cep94A)	*Saccharophagus degradans* 2-40		α-Glc1P	d-glucose	Cellobiose	[73]
GH94	Inverting	ABN51514.1	Cellobiose phosphorylase (Cbp;CtCBP;Cthe_0275)	*Acetivibrio thermocellus* ATCC 27405		α-Glc1P	d-glucose	Cellobiose	[74]
GH94	Inverting	ACL76454.1	Cellobiose phosphorylase (CbpA;Ccel_2109)	*Ruminiclostridium cellulolyticum* H10		α-Glc1P	d-glucose	Cellobiose	[75]
GH94	Inverting	ADU20744.1	Cellobiose phosphorylase (CBP;RaCBP;Rumal_0187)	*Ruminococcus albus* 7 (DSM 20455)		α-Glc1P	d-glucose	Cellobiose	[24]
GH94	Inverting	BAA25846.1	Cellobiose-phosphorylase (Cbp)	*Acetivibrio thermocellus* YM4		α-Glc1P	d-glucose	Cellobiose	[76]
GH94	Inverting	BAA28631.1	Cellobiose phosphorylase (Cbp;CgCBP)	*Cellulomonas gilvus* ATCC13127	2CQS, 2CQT, 3ACS, 3ACT, 3AFJ, 3QFY, 3QFZ, 3QG0	α-Glc1P	d-glucose	Cellobiose	[77]
GH94	Inverting	CAB16926.1*	Cellobiose phosphorylase (CepA)	*Thermotoga neapolitana* Z2706-MC24		α-Glc1P	d-glucose	Cellobiose	[78]
GH94	Inverting	AAC45511.1	Cellodextrin phosphorylase (CepB; CsCdP)	*Thermoclostridium stercorarium* DSM8532		α-Glc1P	[(1→4)-β-d-glucosyl]_n_	[(1→4)-β-d-glucosyl]_n+1_	[69]
GH94	Inverting	ACL76454.1	Cellobiose phosphorylase (CbpA;Ccel_2109)	*Ruminiclostridium cellulolyticum H10*		α-Glc1P	d-glucose	Cellobiose	[29]
GH94	Inverting	QCO92836.1	Cellobiose phosphorylase (GH94_G)	*Uncultured bacterium*		α-Glc1P	d-glucose	Cellobiose	[79]
GH94	Inverting	QCO92799.1	Cellobiose phosphorylase (GH94_E)	*Uncultured bacterium*		α-Glc1P	d-glucose	Cellobiose	[79]
GH94	Inverting	ADU22883.1	Cellodextrin phosphorylase (CDP;RaCDP;Rumal_2403)	*Ruminococcus albus* 7 (DSM 20455)		α-Glc1P	[(1→4)-β-d-glucosyl]_n_	[(1→4)-β-d-glucosyl]_n+1_	[80]
GH94	Inverting	ACD71661.1	Cyclic β-1,2-glucan synthetase (Cgs)	*Brucella abortus* S19		α-Glc1P	[(1→2)-β-d-glucosyl]_n_	[(1→2)-β-d-glucosyl]_n+1_	[81]
GH94	Inverting	ACL75793.1	Cellodextrin phosphorylase (CdpA; Ccel_1439)	*Ruminiclostridium cellulolyticum* H10		α-Glc1P	[(1→4)-β-d-glucosyl]_n_	[(1→4)-β-d-glucosyl]_n+1_	[29]
GH94	Inverting	ACL77700.1	Cellodextrin phosphorylase (CdpC; Ccel_3412)	*Ruminiclostridium cellulolyticum* H10		α-Glc1P	[(1→4)-β-d-glucosyl]_n_	[(1→4)-β-d-glucosyl]_n+1_	[29]
GH94	Inverting	ACL76688.1	Cellodextrin phosphorylase (CdpB; Ccel_2354)	*Ruminiclostridium cellulolyticum* H10		α-Glc1P	[(1→4)-β-d-glucosyl]_n_	[(1→4)-β-d-glucosyl]_n+1_	[29]
GH94	Inverting	QCO92991.1	Cellodextrin phosphorylase (GH94_A)	Uncultured bacterium		α-Glc1P	[(1→4)-β-d-glucosyl]_n_	[(1→4)-β-d-glucosyl]_n+1_	[29]
GH94	Inverting	QCO92946.1	Cellodextrin phosphorylase (GH94_D)	Uncultured bacterium		α-Glc1P	[(1→4)-β-d-glucosyl]_n_	[(1→4)-β-d-glucosyl]_n+1_	[29]
GH94	Inverting	BAJ10826.1	Laminaribiose phosphorylase (LbpA)	*Paenibacillus sp.* YM1	6GGY, 6GH2, 6GH3	α-Glc1P	d-glucose	Laminaribiose: (β-d-glucopyranosyl-(1,3)-d-glucopyranose)	[82,83]
GH94	Inverting	AAG23740.1	Diacetylchitobiose phosphorylase (ChbP)	*Vibrio furnissii*		α-GlcNAc-1P	*N*-acetyl-d-glucosamine	*N*,*N*’-diacetylchitobiose: (*N*-acetyl-d-glucosaminyl-(1,4)-*N*-acetyl-d-glucosamine)	[84]
GH94	Inverting	BAC87867.1	Chitobiose phosphorylase (ChbP;VpChBP)	*Vibrio proteolyticus*	1V7V, 1V7W, 1V7X	α-GlcNAc-1P	*N*-acetyl-d-glucosamine	*N*,*N*’-diacetylchitobiose	[85]
GH94	Inverting	AFH48717.1	Chitobiose phosphorylase (IaGP;IALB_1006)	*Ignavibacterium album* JCM 16511		α-GlcNAc-1P	*N*-acetyl-d-glucosamine	*N,N’*-diacetylchitobiose	[86]
GH94	Inverting	CAC97070.1	β-1,2-oligoglucan phosphorylase (LiSOGP;Lin1839)	*Listeria innocua* Clip11262		α-Glc1P	[(1→2)-β-d-glucosyl]_n_	[(1→2)-β-d-glucosyl]_n+1_	[87]
GH94	Inverting	ABX41081.1	β-1,2-oligoglucan phosphorylase (LpSOGP;Cphy_0694)	*Lachnoclostridium phytofermentans* ISDg	5H3Z, 5H40, 5H41, 5H42	α-Glc1P	[(1→2)-β-d-glucosyl]_n_	[(1→2)-β-d-glucosyl]_n+1_	[88]
GH94	Inverting	QCO92990.1	Laminaribiose phosphorylase (GH94_B)	Uncultured bacterium		α-Glc1P	d-glucose	Laminaribiose	[79]
GH94	Inverting	QCO92945.1	Laminaribiose phosphorylase (GH94_C)	Uncultured bacterium		α-Glc1P	d-glucose	Laminaribiose	[79]
GH94	Inverting	BAB71818.1	Cellodextrin phosphorylase (Cdp-ym4)	*Acetivibrio thermocellus* YM4		α-Glc1P	[(1→4)-β-d-glucosyl]_n_	[(1→4)-β-d-glucosyl]_n+1_	[71]
GH94	Inverting	ABN54185.1	Cellodextrin-phosphorylase (Cdp;CtCDP;Cthe_2989)	*Acetivibrio thermocellus* ATCC 27405	5NZ7, 5NZ8	α-Glc1P	[(1→4)-β-d-glucosyl]_n_	[(1→4)-β-d-glucosyl]_n+1_	[89]
GH94	Inverting	ADZ85667.1*	Cellodextrin phosphorylase (Cdp;ClCDP;Clole_3989)	*Clostridium lentocellum*		α-Glc1P	[(1→4)-β-d-glucosyl]_n_	[(1→4)-β-d-glucosyl]_n+1_	[73]
GH94	Inverting	AAM43298.1	Cellobionic acid phosphorylase (CelAP;NdvB;XCC4077)	*Xanthomonas campestris pv. campestris str*. ATCC 33913		α-Glc1P	d-gluconate	β-d-glucopyranosyl-(1,4)-d-gluconate	[90]
GH94	Inverting	ABD80168.1	Cellobionic acid phosphorylase 94B (Cep94B;CBAP; SdCBAP;Sde_0906)	*Saccharophagus degradans* 2-40	4ZLE, 4ZLF, 4ZLG, 4ZLI	α-Glc1P	d-gluconate	β-d-glucopyranosyl-(1,4)-d-gluconate	[91]
GH94	Inverting	EAA28929.1	Cellobionic acid phosphorylase (CelAP;NdvB; NCU09425)	*Neurospora crassa* OR74A		α-Glc1P	d-gluconate	β-d-glucopyranosyl-(1,4)-d-gluconate	[90]
GH94	Inverting	AAC45511.1	Cellodextrin phosphorylase (CepB;CsCdP)	*Thermoclostridium stercorarium* DSM8532		α-Glc1P	[(1→4)-β-d-glucosyl]_n_	[(1→4)-β-d-glucosyl]_n+1_	[69]
GH94	Inverting	AHC20114.1	Glucosylgalactose Phosphorylase (GGalP)	*Paenibacillus polymyxa* CR1		α-Glc1P	d-galactose	β-d-glucopyranosyl-(1,4)-d-galactose	[86]
GH94	Inverting	ABX81345.1	Laminaribiose phosphorylase (ACL_0729;ACL0729)	*Acholeplasma laidlawii* PG-8A		α-Glc1P	d-glucose	Laminaribiose	[92]
GH112	Inverting	ACB74662.1	D-galactosyl-β-1,4-L-rhamnose phosphorylase (GalRhaP;Oter_1377)	*Opitutus terrae* PB90-1		α-Gal1P	l-rhamnose	(β-d-galactosyl-(1,4)-l-rhamnose)	[93]
GH112	Inverting	ABX42289.1	D-galactosyl-β-1,4-L-rhamnose phosphorylase (Cphy_1920)	*Lachnoclostridium phytofermentans* ISDg		α-Gal1P	l-rhamnose	(β-d-galactosyl-(1,4)-l-rhamnose)	[94]
GH112	Inverting	AAO07997.1	β-1,3-galactosyl-*N*-acetylhexosamine phosphorylase (GalGlyNAcP;VV2_1091)	*Vibrio vulnificus* CMCP6		α-Gal1P	*N*-acetyl-d-glucosamine	(β-d-galactosyl-(1,3)-*N*-acetyl-d-glucosamine)	[95]
GH112	Inverting	ABX40964.1	β-1,3-galactosyl-*N*-acetylhexosamine phosphorylase (Cphy_0577)	*Lachnoclostridium phytofermentans* ISDg		α-Gal1P	*N*-acetyl-d-glucosamine *N*-acetyl-d-galactosamine	(β-d-galactosyl-(1,3)-*N*-acetyl-d-glucosamine) (β-d-galactosyl-(1,3)-*N*-acetyl-d-galactosamine)	[94]
GH112	Inverting	ABX43387.1	β-1,3-galactosyl-*N*-acetylhexosamine phosphorylase (Cphy_3030)	*Lachnoclostridium phytofermentans* ISDg		α-Gal1P	*N*-acetyl-d-glucosamine *N*-acetyl-d-galactosamine	(β-d-galactosyl-(1,3)-*N*-acetyl-d-glucosamine) (β-d-galactosyl-(1,3)-*N*-acetyl-d-galactosamine)	[94]
GH112	Inverting	AEN95946.1	β-1,3-galactosyl-*N*-acetylhexosamine phosphorylase (RhGLnbp112;RHOM_04120)	*Roseburia hominis* A2-183		α-Gal1P	*N*-acetyl-d-glucosamine *N*-acetyl-d-galactosamine	(β-d-galactosyl-(1,3)-*N*-acetyl-d-glucosamine) (β-d-galactosyl-(1,3)-*N*-acetyl-d-galactosamine)	[96]
GH112	Inverting	EEG94248.1	β-1,3-galactosyl-*N*-acetylhexosamine phosphorylase (RiGLnbp112; ROSEINA2194_01886; GnpA)	*Roseburia inulinivorans* DSM 16841		α-Gal1P	*N*-acetyl-d-glucosamine *N*-acetyl-d-galactosamine	(β-d-galactosyl-(1,3)-*N*-acetyl-d-glucosamine) (β-d-galactosyl-(1,3)-*N*-acetyl-d-galactosamine)	[96]
GH112	Inverting	BAD80751.1	Galacto-*N*-biose/lacto-*N*-biose phosphorylase (LnpA1;LnbP;GLNBP; BLLJ_1623)	*Bifidobacterium longum subsp. longum* JCM 1217	2ZUS, 2ZUT, 2ZUU, 2ZUV, 2ZUW, 3WFZ	α-Gal1P	*N*-acetyl-d-glucosamine *N*-acetyl-d-galactosamine	(β-d-galactosyl-(1,3)-*N*-acetyl-d-glucosamine) (β-d-galactosyl-(1,3)-*N*-acetyl-d-galactosamine)	[97]
GH112	Inverting	BAD80752.1	Lacto-*N*-biose phosphorylase (LnpA1;LnbP)	*Bifidobacterium bifidum* JCM 1254		α-Gal1P	*N*-acetyl-d-glucosamine *N*-acetyl-d-galactosamine	(β-d-galactosyl-(1,3)-*N*-acetyl-d-glucosamine) (β-d-galactosyl-(1,3)-*N*-acetyl-d-galactosamine)	[98]
GH112	Inverting	ACV29689.1	β-1,3-galactosyl-*N*-acetylhexosamine phosphorylase/galacto-*N*-biose/lacto-*N*-biose I phosphorylase (GLNBP;Apre_1669)	*Anaerococcus prevotii* DSM 20548		α-Gal1P	*N*-acetyl-d-glucosamine *N*-acetyl-d-galactosamine	(β-d-galactosyl-(1,3)-*N*-acetyl-d-glucosamine) (β-d-galactosyl-(1,3)-*N*-acetyl-d-galactosamine)	[26]
GH112	Inverting	ZP_05748149.1	β-1,3-galactosyl-*N*-acetylhexosamine phosphorylase/galacto-*N*-biose phosphorylase (GNBP;HMPREF0357_1319)	*Erysipelothrix rhusiopathiae* ATCC 19414		α-Gal1P	*N*-acetyl-d-galactosamine	(β-d-galactosyl-(1,3)-*N*-acetyl-d-galactosamine)	[26]
GH112	Inverting	ACZ00636.1	β-1,3-galactosyl-*N*-acetylhexosamine phosphorylase/galacto-*N*-biose phosphorylase (GNBP;Smon_0146)	*Streptobacillus moniliformis* DSM 12112		α-Gal1P	*N*-acetyl-d-galactosamine	(β-d-galactosyl-(1,3)-*N*-acetyl-d-galactosamine)	[26]
GH112	Inverting	ABG83511.1	Galacto-*N*-biose phosphorylase (CPF_0553)	*Clostridium perfringens* ATCC 13124		α-Gal1P	*N*-acetyl-d-galactosamine	(β-d-galactosyl-(1,3)-*N*-acetyl-d-galactosamine)	[95]
GH112	Inverting	BAH10636.1	β-1,3-galactosyl-*N*-acetylhexosamine phosphorylase (GnpA)	*Cutibacterium acnes* JCM 6425		α-Gal1P	*N*-acetyl-d-galactosamine	(β-d-galactosyl-(1,3)-*N*-acetyl-d-galactosamine)	[99]
GH130	Inverting	CAC96089.1	β-1,2-mannobiose phosphorylase (Lin0857)	*Listeria innocua* Clip11262	5B0P, 5B0Q, 5B0R, 5B0S	α-Man1P	d-mannose	β-1,2-mannobiose	[100]
GH130	Inverting	ABY93074.1	β-1,2-mannobiose phosphorylase (Teth514_1789)	*Thermoanaerobacter sp*. X514	7FIP, 7FIQ, 7FIR, 7FIS	α-Man1P	d-mannose	β-1,2-mannobiose	[101]
GH130	Inverting	ABY93073.1	β-1,2-oligomannan phosphorylase (Teth514_1788)	*Thermoanaerobacter sp.* X514		α-Man1P	(β-1,2-d-mannose)_n_	(β-1,2-d-mannose)_n+1_	[101]
GH130	Inverting	CAZ94304.1	β-1,3-mannooligosaccharide phosphorylase (zobellia_231)	*Zobellia galactanivorans* DsijT		α-Man1P	(β-1,3-d-mannose)_n_	(β-1,3-d-mannose)_n+1_	[102]
GH130	Inverting	AAS19693.1	β-1,4-mannosylglucose phosphorylase (UnkA, CvMGP)	*Cellvibrio vulgaris* NCIMB8633		α-Man1P	d-glucose	β-mannopyranosyl-(1→4)-d-glucopyranose	[103]
GH130	Inverting	ADU21379.1	β-1,4-mannosylglucose phosphorylase (RaMP1; RaMGP; Rumal_0852)	*Ruminococcus albus* 7 (DSM 20455)	5AY9, 5AYC	α-Man1P	d-glucose	β-mannopyranosyl-(1→4)-d-glucopyranose	[104]
GH130	Inverting	CAH06518.1	β-1,4-mannosylglucose phosphorylase (MGP; BF0772)	*Bacteroides fragilis* NCTC 9343	3WAS, 3WAT, 3WAU, 4KMI	α-Man1P	d-glucose	β-mannopyranosyl-(1→4)-d-glucopyranose	[105]
GH130	Inverting	VCV21228.1	β-1,4-mannosylglucose phosphorylase (RIL182_01099; ROSINTL182_07685)	*Roseburia intestinalis* L1-82		α-Man1P	d-glucose	β-mannopyranosyl-(1→4)-d-glucopyranose	[106]
GH130	Inverting	ACY49318.1	β-1,4-mannosylglucose phosphorylase (MGP; RmMGP; Rmar_2440)	*Rhodothermus marinus* R-10T (ATCC43812)		α-Man1P	d-glucose	β-mannopyranosyl-(1→4)-d-glucopyranose	[107]
GH130	Inverting	AAO76140.1	β-1,4-mannosyl-*N*-acetyl-glucosamine phosphorylase (BT1033)	*Bacteroides thetaiotaomicron* VPI-5482		α-Man1P	*N*-acetyl-d-glucosamine	β-mannopyranosyl-(1→4)-*N*-acetyl-d-glucosamine	[108]
GH130	Inverting	ADD61463.1	β-1,4-mannopyranosyl-[*N*-glycan] phosphorylase β-1,4-mannopyranosyl-chitobiose phosphorylase (Uhgb_MP)	Uncultured bacterium	4UDG, 4UDI, 4UDJ, 4UDK	α-Man1P	β-*N*-acetyl-d-glucosaminyl-(1,4)-*N*-acetyl-d-glucosamine	β-1,4-mannopyranosyl-chitobiose (β-mannopyranosyl-(1→4)-*N*-acetyl-β-d-glucosaminyl-(1→4)-*N*-acetyl-d-glucosamine)	[109]
GH130	Inverting	ADU20661.1	β-1,4-mannooligosaccharide phosphorylase (MOP; RaMP2; Rumal_0099)	*Ruminococcus albus* 7 (DSM 20455)	5AYD, 5AYE	α-Man1P	(β-1,4-d-mannose)_n_	(β-1,4-d-mannose)_n+1_	[104]
GH130	Inverting	VCV21229.1	β-1,4-mannooligosaccharide phosphorylase (RIL182_01100; ROSINTL182_05474)	*Roseburia intestinalis* L1-82		α-Man1P	(β-1,4-d-mannose)_n_	(β-1,4-d-mannose)_n+1_	[106]
GH130	Inverting	AAD36300.1	β-1,4-mannooligosaccharide phosphorylase (TM1225)	*Thermotoga maritima* MSB8	1VKD	α-Man1P	(β-1,4-d-mannose)_n_	(β-1,4-d-mannose)_n+1_	[110]
GH130	Inverting	WP_026485574.1	β-1,4-mannooligosaccharide phosphorylase (CalpoDRAFT_0075)	*Caldanaerobius polysaccharolyticus* ATCC BAA-17		α-Man1P	(β-1,4-d-mannose)_n_	(β-1,4-d-mannose)_n+1_	[111]
GH130	Inverting	WP_026486530.1	β-1,4-mannooligosaccharide phosphorylase (CalpoDRAFT_1209)	*Caldanaerobius polysaccharolyticus* ATCC BAA-17		α-Man1P	(β-1,4-d-mannose)_n_	(β-1,4-d-mannose)_n+1_	[111]
GH130	Inverting	CAH1385115.1	β-1,4-mannosyl-glucuronic acid phosphorylase	Uncultured bacterium		α-Man1P	d-glucuronic acid	β-mannopyranosyl-(1→4)-d-glucuronic acid	[112]
GH130	Inverting	CAH1385116.1	β-1,4-mannooligosaccharide phosphorylase	Uncultured bacterium		α-Man1P	d-mannose β-1,4-mannobiose	β-1,4-mannobiose β-1,4-mannotriose	[112]
GH130	Inverting	CAH1385117.1	β-1,3-mannosyl-glucose phosphorylase β-1,3-mannooligosaccharide phosphorylase	Uncultured bacterium		α-Man1P	d-glucose d-mannose	β-mannopyranosyl-(1→3)-d-glucopyranose β-1,3-mannobiose	[112]
GH130	Inverting	CAH1385118.1	Mixed-linkage mannoside synthase Uhgb_MS	Uncultured bacterium		α-Man1P	d-mannose α-1,2-mannobiose	β-1,2-mannobiose α-1,2-mannotriose β-d-mannopyranosyl-(1→2)-α-1,2-mannotriose	[113]
GH149	Inverting		β-1,3-oligoglucan phosphorylase (Pro_7066)	Uncultured bacterium	6HQ6, 6HQ8, 6HQ6	α-Glc1P	[(1→3)-β-d-glucosyl]_n_	[(1→3)-β-d-glucosyl]_n+1_	[114]
GH149	Inverting	AUO30192.1	β-1,3-glucan phosphorylase (EgP1)	Euglena gracilis		α-Glc1P	[(1→3)-β-d-glucosyl]_n_	[(1→3)-β-d-glucosyl]_n+1_	[115]
GH149	Inverting	QCO92814.1	β-1,3-glucan phosphorylase (GH149_H)	Uncultured bacterium		α-Glc1P	[(1→3)-β-d-glucosyl]_n_	[(1→3)-β-d-glucosyl]_n+1_	[79]
GH161	Inverting	WP_019688419.1	β-1,3-glucan phosphorylase (PPT_RS0121460; PapP)	*Paenibacillus polymyxa* ATCC 842		α-Glc1P	[(1→3)-β-d-glucosyl]_n_	[(1→3)-β-d-glucosyl]_n+1_	[116]
GH161	Inverting	ACJ76363.1	β-1,3-glucan phosphorylase (THA_1941; TaCDP)	*Thermosipho africanus* TCF52B		α-Glc1P	[(1→3)-β-d-glucosyl]_n_	[(1→3)-β-d-glucosyl]_n+1_	[116]
GH161	Inverting	BAU78234.1	β-1,3-glucan phosphorylase (OdBGP)	*Ochromonas danica* NIES-2142		α-Glc1P	[(1→3)-β-d-glucosyl]_n_	[(1→3)-β-d-glucosyl]_n+1_	[117]
GT4	Retaining	BAA31350.1	Trehalose synthase (Tsase)	*Grifola frondosa*		α-Glc1P	d-glucose	Trehalose: (α-d-glucosyl-(1,1)-d-glucose)	[118]
GT4	Retaining	AAF22230.1	Threalose phosphorylase (PsTP)	*Lentinus sajor-caju* ASI2070 (*Pleurotus sajor-caju*)		α-Glc1P	d-glucose	trehalose	[119]
GT4	Retaining	ABC84380.1	Threalose phosphorylase	*Schizophyllum commune* BT2115		α-Glc1P	d-glucose	trehalose	[120,121]
GT35	Retaining	ABP51432.1	Glucan/maltodextrin phosphorylase (GlgP; PyglgP; Pars_1881)	*Pyrobaculum arsenaticum* DSM 13514		α-Glc1P	[(1→4)-α-d-glucosyl]_n_	[(1→4)-α-d-glucosyl]_n+1_	[122]
GT35	Retaining	AAL81659.1	α-glucan/maltodextrin phosphorylase (PF1535)	*Pyrococcus furiosus* DSM 3638		α-Glc1P	[(1→4)-α-d-glucosyl]_n_	[(1→4)-α-d-glucosyl]_n+1_	[123,124]
GT35	Retaining	AAD28735.1	Maltodextrin phosphorylase (MalP)	*Thermococcus litoralis*		α-Glc1P	[(1→4)-α-d-glucosyl]_n_	[(1→4)-α-d-glucosyl]_n+1_	[125]
GT35	Retaining	ABN51595.1	α-glucan phosphorylase (α-GP; Cthe_0357)	*Acetivibrio thermocellus* ATCC 27405		α-Glc1P	[(1→4)-α-d-glucosyl]_n_	[(1→4)-α-d-glucosyl]_n+1_	[126]
GT35	Retaining	CAB61943.1	α-glucan phosphorylase (PHS2; AtPHS2; At3g46970)	*Arbidopsis thaliana*		α-Glc1P	[(1→4)-α-d-glucosyl]_n_	[(1→4)-α-d-glucosyl]_n+1_	[127]
GT35	Retaining	AAD03471.1	Glycogen phosphorylase (GlgP)	*Agrobacterium tumefaciens* A348		α-Glc1P	[(1→4)-α-d-glucosyl]_n_	[(1→4)-α-d-glucosyl]_n+1_	[128]
GT35	Retaining	AAC06896.1	Glycogen phosphorylase (GlgP; Aq_717)	*Aquifex aeolicus* VF5		α-Glc1P	[(1→4)-α-d-glucosyl]_n_	[(1→4)-α-d-glucosyl]_n+1_	[129]
GT35	Retaining	AAC00218.1	Glycogen phosphorylase (GlgP; BSU30940)	*Bacillus subtilis subsp. subtilis* str. 168		α-Glc1P	[(1→4)-α-d-glucosyl]_n_	[(1→4)-α-d-glucosyl]_n+1_	[130]
GT35	Retaining	AAM24997.1	α-glucan phosphorylase (GlgP; TTE1805)	*Caldanaerobacter subterraneus subsp. tengcongensis* MB4		α-Glc1P	[(1→4)-α-d-glucosyl]_n_	[(1→4)-α-d-glucosyl]_n+1_	[131]
GT35	Retaining	AAM52219.1	Glycogen phosphorylase	*Corynebacterium callunae*	2C4M	α-Glc1P	[(1→4)-α-d-glucosyl]_n_	[(1→4)-α-d-glucosyl]_n+1_	[132]
GT35	Retaining	BAB98701.1	Maltodextrin/glycogen phosphorylase (MalP; GlgP1; cg1479)	*Corynebacterium glutamicum* ATCC 13032		α-Glc1P	[(1→4)-α-d-glucosyl]_n_	[(1→4)-α-d-glucosyl]_n+1_	[133]
GT35	Retaining	BAB99480.1	Glycogen phosphorylase (GlgP; GlgP2; cg2289)	*Corynebacterium glutamicum* ATCC 13032		α-Glc1P	[(1→4)-α-d-glucosyl]_n_	[(1→4)-α-d-glucosyl]_n+1_	[133]
GT35	Retaining	AAC76453.1	Glycogen phosphorylase (GlgP; b3428)	*Escherichia coli str. K-12 substr.* MG1655		α-Glc1P	[(1→4)-α-d-glucosyl]_n_	[(1→4)-α-d-glucosyl]_n+1_	[134]
GT35	Retaining	AAC76442.1	Maltodextrin phosphorylase (MalP; b3417)	*Escherichia coli str. K-12 substr*. MG1655	1AHP, 1E4O, 1L5V, 1L5W, 1L6I, 1QM5, 2ASV, 2AV6, 2AW3, 2AZD, 2ECP	α-Glc1P	[(1→4)-α-d-glucosyl]_n_	[(1→4)-α-d-glucosyl]_n+1_	[135] [136]
GT35	Retaining	BAA19592.1	Glycogen phosphorylase (GlgP)	*Geobacillus stearothermophilus* TRBE14		α-Glc1P	[(1→4)-α-d-glucosyl]_n_	[(1→4)-α-d-glucosyl]_n+1_	[137]
GT35	Retaining	AAN59210.1	Maltodextrin phosphorylase (GlgP; SMU.1564)	*Streptococcus mutans* UA159	4L22	α-Glc1P	[(1→4)-α-d-glucosyl]_n_	[(1→4)-α-d-glucosyl]_n+1_	[138]
GT35	Retaining	AAL26558.1	Glycogen phosphorylase (Glg)	*Synechococcus elongatus* PCC 7942 = FACHB-805		α-Glc1P	[(1→4)-α-d-glucosyl]_n_	[(1→4)-α-d-glucosyl]_n+1_	[139]
GT35	Retaining	AGL50099.1	α-glucan phosphorylase (AgpA; Tmari_1175)	*Thermotoga maritima* MSB8		α-Glc1P	[(1→4)-α-d-glucosyl]_n_	[(1→4)-α-d-glucosyl]_n+1_	[140]
GT35	Retaining	BAB11741.1	α-glucan phosphorylase (GlgP)	*Thermus aquaticus*		α-Glc1P	[(1→4)-α-d-glucosyl]_n_	[(1→4)-α-d-glucosyl]_n+1_	[141]
GT35	Retaining	CAC93400.1	Glycogen phosphorylase (glgP, YPO3938)	*Yersinia pestis* CO92		α-Glc1P	[(1→4)-α-d-glucosyl]_n_	[(1→4)-α-d-glucosyl]_n+1_	[142]
GT35	Retaining	AAB46846.1	Glycogen phosphorylase (myophosphorylase)	*Bos taurus*		α-Glc1P	[(1→4)-α-d-glucosyl]_n_	[(1→4)-α-d-glucosyl]_n+1_	[143]
GT35	Retaining	ABB88567.1	Plastidial starch phosphorylase (PhoB; Sta4)	*Chlamydomonas reinhardtii* 137C		α-Glc1P	[(1→4)-α-d-glucosyl]_n_	[(1→4)-α-d-glucosyl]_n+1_	[144]
GT35	Retaining	CAA44069.1	Glycogen phosphorylase 1 (GlpV; GP1)	*Dictyostelium discoideum*		α-Glc1P	[(1→4)-α-d-glucosyl]_n_	[(1→4)-α-d-glucosyl]_n+1_	[145]
GT35	Retaining	AAA33211.1	Glycogen phosphorylase 2 (GlpD; GP2)	*Dictyostelium discoideum*		α-Glc1P	[(1→4)-α-d-glucosyl]_n_	[(1→4)-α-d-glucosyl]_n+1_	[146]
GT35	Retaining	AAD46887.1	glycogen phosphorylase (GlyP; GLYP; CG7254; Dmel_CG7254)	*Drosophila melanogaster*		α-Glc1P	[(1→4)-α-d-glucosyl]_n_	[(1→4)-α-d-glucosyl]_n+1_	[147]
GT35	Retaining	AAN17338.1	Glycogen phosphorylase-2	*Entamoeba histolytica*		α-Glc1P	[(1→4)-α-d-glucosyl]_n_	[(1→4)-α-d-glucosyl]_n+1_	[148]
GT35	Retaining	AAL23578.1	Glycogen phosphorylase	*Entamoeba histolytica*		α-Glc1P	[(1→4)-α-d-glucosyl]_n_	[(1→4)-α-d-glucosyl]_n+1_	[148]
GT35	Retaining	AAP33020.1	Glycogen phosphorylase	*Gallus gallu*s		α-Glc1P	[(1→4)-α-d-glucosyl]_n_	[(1→4)-α-d-glucosyl]_n+1_	[149]
GT35	Retaining	AAK69600.1	Glycogen phosphorylase	*Giardia intestinalis* ATCC30957		α-Glc1P	[(1→4)-α-d-glucosyl]_n_	[(1→4)-α-d-glucosyl]_n+1_	[148]
GT35	Retaining	AAB60395.1	Glycogen phosphorylase (brain) (bGP)	*Homo sapiens*	5IKO, 5IKP	α-Glc1P	[(1→4)-α-d-glucosyl]_n_	[(1→4)-α-d-glucosyl]_n+1_	[150]
GT35	Retaining	CAA75517.1	Glycogen phosphorylase (liver)	*Homo sapiens*	1EM6, 1EXV, 1FA9, 1FC0, 1L5Q, 1L5R, 1L5S, 1L7X, 1XOI, 2ATI, 2QLL, 2ZB2, 3CEH, 3CEJ, 3CEM, 3DD1, 3DDS, 3DDW	α-Glc1P	[(1→4)-α-d-glucosyl]_n_	[(1→4)-α-d-glucosyl]_n+1_	[151]
GT35	Retaining	AAC17451.1	Glycogen phosphorylase (muscle) (PigM)	*Homo sapiens*	1Z8D	α-Glc1P	[(1→4)-α-d-glucosyl]_n_	[(1→4)-α-d-glucosyl]_n+1_	[152]
GT35	Retaining	BAK00834.1	Plastidial α-1,4-glucan phosphorylase (Pho1; HvPho1)	*Hordeum vulgare subsp. Vulgare*	5LR8, 5LRA, 5LRB	α-Glc1P	[(1→4)-α-d-glucosyl]_n_	[(1→4)-α-d-glucosyl]_n+1_	[153]
GT35	Retaining	AAA63271.1	α-glucan phosphorylase L	*Ipomoea batatas*		α-Glc1P	[(1→4)-α-d-glucosyl]_n_	[(1→4)-α-d-glucosyl]_n+1_	[154]
GT35	Retaining	AAK01137.1	Starch phosphorylase (fragment)	*Ipomoea batatas*		α-Glc1P	[(1→4)-α-d-glucosyl]_n_	[(1→4)-α-d-glucosyl]_n+1_	[155]
GT35	Retaining	AAL23577.1	Glycogen phosphorylase	*Mastigamoeba balamuthi*		α-Glc1P	[(1→4)-α-d-glucosyl]_n_	[(1→4)-α-d-glucosyl]_n+1_	[148]
GT35	Retaining	AAD30476.1	Glycogen phosphorylase (muscle) (PygM)	*Mus musculus*		α-Glc1P	[(1→4)-α-d-glucosyl]_n_	[(1→4)-α-d-glucosyl]_n+1_	[156]
GT35	Retaining	AAG00588.1	Glycogen phosphorylase	*Mus musculus*		α-Glc1P	[(1→4)-α-d-glucosyl]_n_	[(1→4)-α-d-glucosyl]_n+1_	[157]
GT35	Retaining	ACJ76617.1	Glycogen phosphorylase (muscle) (PygM)	*Oryctolagus cuniculus*	Dozens available—2GJ4[A] best solution	α-Glc1P	[(1→4)-α-d-glucosyl]_n_	[(1→4)-α-d-glucosyl]_n+1_	[158,159]
GT35	Retaining	AAK15695.1	α-1,4-glucan phosphorylase L	*Oryza sativa*		α-Glc1P	[(1→4)-α-d-glucosyl]_n_	[(1→4)-α-d-glucosyl]_n+1_	[160]
GT35	Retaining	BAB92854.1	α-1,4-glucan phosphorylase (Os01g0851700)	*Oryza sativa Japonica* Group		α-Glc1P	[(1→4)-α-d-glucosyl]_n_	[(1→4)-α-d-glucosyl]_n+1_	[161]
GT35	Retaining	AAV87308.1	brain glycogen phosphorylase (PYGB)	*Ovis aries*		α-Glc1P	[(1→4)-α-d-glucosyl]_n_	[(1→4)-α-d-glucosyl]_n+1_	[162]
GT35	Retaining	AAB68800.1	glycogen phosphorylase (muscle)	*Ovis aries*		α-Glc1P	[(1→4)-α-d-glucosyl]_n_	[(1→4)-α-d-glucosyl]_n+1_	[163]
GT35	Retaining	AAA41252.1	glycogen phosphorylase (Brain)	*Rattus norvegicus*		α-Glc1P	[(1→4)-α-d-glucosyl]_n_	[(1→4)-α-d-glucosyl]_n+1_	[164]
GT35	Retaining	AAH70901.1	glycogen phosphorylase (liver)	*Rattus norvegicus*		α-Glc1P	[(1→4)-α-d-glucosyl]_n_	[(1→4)-α-d-glucosyl]_n+1_	[165]
GT35	Retaining	AAA41253.1	glycogen phosphorylase (muscle)	*Rattus norvegicus*		α-Glc1P	[(1→4)-α-d-glucosyl]_n_	[(1→4)-α-d-glucosyl]_n+1_	[166]
GT35	Retaining	AAB68057.1	Glycogen phosphorylase (Gph1; YPR160w)	*Saccharomyces cerevisiae* S288c	1YGP	α-Glc1P	[(1→4)-α-d-glucosyl]_n_	[(1→4)-α-d-glucosyl]_n+1_	[167,168]
GT35	Retaining	AAA33809.1	α-glucan phosphorylase H	*Solanum tuberosum*		α-Glc1P	[(1→4)-α-d-glucosyl]_n_	[(1→4)-α-d-glucosyl]_n+1_	[169]
GT35	Retaining	BAA00407.1	α-glucan phosphorylase L1	*Solanum tuberosum*		α-Glc1P	[(1→4)-α-d-glucosyl]_n_	[(1→4)-α-d-glucosyl]_n+1_	[170]
GT35	Retaining	CAA52036.1	α-glucan phosphorylase L2	*Solanum tuberosum*		α-Glc1P	[(1→4)-α-d-glucosyl]_n_	[(1→4)-α-d-glucosyl]_n+1_	[171]
GT35	Retaining	CAA59464.1	α-glucan phosphorylase	*Spinacia oleracea*		α-Glc1P	[(1→4)-α-d-glucosyl]_n_	[(1→4)-α-d-glucosyl]_n+1_	[159]
GT35	Retaining	AAL23579.1	glycogen phosphorylase	*Trichomonas vaginalis*		α-Glc1P	[(1→4)-α-d-glucosyl]_n_	[(1→4)-α-d-glucosyl]_n+1_	[148]
GT35	Retaining	AAF82787.1	α-glucan phosphorylase	*Triticum aestivum*		α-Glc1P	[(1→4)-α-d-glucosyl]_n_	[(1→4)-α-d-glucosyl]_n+1_	[172]
GT35	Retaining	CAA84494.1	α-glucan phosphorylase (Pho2;VfPho2)	*Vicia faba var. minor*		α-Glc1P	[(1→4)-α-d-glucosyl]_n_	[(1→4)-α-d-glucosyl]_n+1_	[173]
GT35	Retaining	CAA85354.1	α-glucan phosphorylase L	*Vicia faba var. minor*		α-Glc1P	[(1→4)-α-d-glucosyl]_n_	[(1→4)-α-d-glucosyl]_n+1_	[173]
GT108	Inverting	CBZ24448.1	MTP3 (LMXM_10_1250)	*Leishmania mexicana* MHOM/GT/2001/U1103		α-Man1P	(β-1,2-d-mannose)_n_	(β-1,2-d-mannose)_n+1_	[174]
GT108	Inverting	CBZ24451.1	MTP6 (LMXM_10_1280)	*Leishmania mexicana* MHOM/GT/2001/U1103		α-Man1P	(β-1,2-d-mannose)_n_	(β-1,2-d-mannose)_n+1_	[174]
GT108	Inverting	CBZ24449.1	MTP4 (LMXM_10_1260)	*Leishmania mexicana* MHOM/GT/2001/U1103	6Q50	α-Man1P	(β-1,2-d-mannose)_n_	(β-1,2-d-mannose)_n+1_	[174]
GT108	Inverting	CBZ24452.1	MTP7 (LMXM_10_1290)	*Leishmania mexicana* MHOM/GT/2001/U1103		α-Man1P	(β-1,2-d-mannose)_n_	(β-1,2-d-mannose)_n+1_	[174]

**Table 2 ijms-23-03043-t002:** Structurally characterized GPs. Only the enzymes proven to have GP activity are listed. ACJ76617.1 has dozens of PDBs, only the PDB with the best solution is listed.

CAZy Family	GenBank/ UniProtKB Accession	Organism Name	PDB Accession Numbers	Protein Name	Fold	Quaternary Structure	Reference
GH3	AUG44408.1	Uncultured bacterium	5VQD[A], 5VQE[A]	β-glucoside phosphorylase (BglP)	(β/α)8	Monomer	[30]
GH13_18	AAO33821.1	*Bifidobacterium adolescentis* DSM 20083	1R7A[A,B], 2GDU[A,B], 2GDV[A,B], 5C8B[B], 5M9X[B], 5MAN[B], 5MB2[B], 6FME[A,B]	Sucrose phosphorylase (SucP; SP; BaSP)	(β/α)8	Dimer	[196]
GH13_18	BAN03569.1	*Ilumatobacter coccineus* YM16-304	6S9U[A]	Sucrose 6(F)-phosphate phosphorylase (YM304_32550)	(β/α)8	Monomer	[50]
GH13_18	ADL69407.1	*Thermoanaerobacterium thermosaccharolyticum* DSM 571	6S9V[A,B]	Sucrose 6(F)-phosphate phosphorylase (SPP; TtSPP; Tthe_1921)	(β/α)8	Monomer	[50]
GH65	Q7SIE1	*Levilactobacillus brevis* ATCC 8287	1H54[A]	Maltose phosphorylase	(α/α)6	Dimer	[197]
GH65	ADI00307.1	*Bacillus selenitireducens* MLS10	4KTP[A,B], 4KTR[A,B,C,D,E,F,G,H]	2-O-α-glucopyranosylglycerol: phosphate β-glucosyltransferase/ 2-O-α-glucosylglycerol phosphorylase (GGP; Bsel_2816)	(α/α)6	Dimer	[198]
GH65	ABP66077.1	*Caldicellulosiruptor saccharolyticus* DSM 8903	3WIQ[A], 3WIR[A,B,C,D]	Kojibiose phosphorylase (CsKP; Csac_0439	(α/α)6	Dimer	[199]
GH65	/	Uncultured bacterium	6W0P[B,C,D], 6W0P[A]	Kojibiose phosphorylase	(α/α)6	Dimer	Unpublished
GH65	BAB97299.1	*Thermoanaerobacter brockii* ATCC 35047		Trehalose phosphorylase (TreP)	(α/α)6	Dimer	[200]
GH94	ABN54185.1	*Acetivibrio thermocellus* ATCC 27405	5NZ7[A,B], 5NZ8[A,B]	Cellodextrin-phosphorylase (Cdp; CtCDP; Cthe_2989)	(α/α)6	Dimer	[201]
GH94	AAL67138.1	*Acetivibrio thermocellus* YM4	3QDE[A,B]	Cellobiose phosphorylase (Cbp)	(α/α)6	Dimer	[202]
GH94	BAA28631.1	*Cellulomonas gilvus* ATCC13127	2CQS[A,B], 2CQT[A,B], 3ACS[A,B], 3ACT[A,B], 3AFJ[A,B], 3QFY[A,B], 3QFZ[A,B], 3QG0[A,B]	Cellobiose phosphorylase (Cbp; CgCBP)	(α/α)6	Dimer	[203,204]
GH94	AAQ20920.1	*Cellulomonas uda* DSM 20108	3RRS[A,B], 3RSY[A,B], 3S4A[A,B], 3S4B[A,B], 3S4C[A], 3S4D[A]	Cellobiose phosphorylase (CbP)	(α/α)6	Dimer	Unpublished
GH94	ABX41081.1	*Lachnoclostridium phytofermentans* ISDg	5H3Z[A,B], 5H40[A,B], 5H41[A,B], 5H42[A,B]	β-1,2-oligoglucan phosphorylase (LpSOGP; Cphy_0694)	(α/α)6	Monomer	[88]
GH94	BAJ10826.1	*Paenibacillus sp*. YM1	6GGY[A,B], 6GH2[A,B], 6GH3[A,B]	Laminaribiose phosphorylase (LbpA)	(α/α)6	Dimer	[116]
GH94	ABD80168.1	*Saccharophagus degradans* 2-40	4ZLE[A], 4ZLF[A], 4ZLG[A], 4ZLI[A]	Cellobionic acid phosphorylase 94B (Cep94B; CBAP; SdCBAP; Sde_0906)	(α/α)6	Dimer	[91]
GH94	BAC87867.1	*Vibrio proteolyticus*	1V7V[A], 1V7W[A], 1V7X[A]	Chitobiose phosphorylase (ChbP; VpChBP)	(α/α)6	Dimer	[205]
GH112	BAD80751.1	*Bifidobacterium longum subsp. longum* JCM 1217	2ZUS[A,B,C,D], 2ZUT[A,B,C,D], 2ZUU[A,B,C,D], 2ZUV[A,B], 2ZUW[A,B,C,D], 3WFZ[A,B,C,D]	Galacto-*N*-biose phosphorylase/Lacto-*N*-biose phosphorylase (LnpA1; LnbP; GLNBP; BLLJ_1623)	(β/α)8	Dimer	[206,207]
GH130	CAH06518.1	*Bacteroides fragilis* NCTC 9343	3WAS[A,B], 3WAT[A,B], 3WAU[A,B], 4KMI[A,B	β-1,4-mannosylglucose phosphorylase (MGP; BF0772)	5-fold β-propeller	Hexamer	[208]
GH130	CAC96089.1	*Listeria innocua* Clip11262	5B0P[A,B], 5B0Q[A,B], 5B0R[A,B], 5B0S[A,B]	β-1,2-mannobiose phosphorylase (Lin0857)	5-fold β-propeller	Dimer	[100]
GH130	ADU21379.1	*Ruminococcus albus* 7 (DSM 20455)	5AY9[A], 5AYC[A]	β-1,4-mannosylglucose phosphorylase (RaMP1; RaMGP; Rumal_0852)	5-fold β-propeller	Trimer	[209]
GH130	ADU20661.1	*Ruminococcus albus* 7 (DSM 20455)	5AYD[A,B,C,D,E,F], 5AYE[A,B,C,D,E,F]	β-1,4-mannooligosaccharide phosphorylase (MOP; RaMP2; Rumal_0099)	5-fold β-propeller	Hexamer	[209]
GH130	ABY93074.1	*Thermoanaerobacter sp*. X514	7FIP, 7FIQ, 7FIR, 7FIS	β-1,2-mannobiose phosphorylase (Teth514_1789)	5-fold β-propeller	Monomer	[210]
GH130	AAD36300.1	*Thermotoga maritima* MSB8	1VKD[A,B,C,D,E,F]	β-1,4-mannooligosaccharide phosphorylase (TM1225)	5-fold β-propeller	Dimer	Unpublished
GH130	ADD61463.1	Uncultured bacterium	4UDG[A,B,C,D,E,F], 4UDI[A,B,C,D,E,F], 4UDJ[A,B,C,D,E,F], 4UDK[A,B,C,D,E,F]	β-1,4-mannopyranosyl-[*N*-glycan] phosphorylase/β-1,4-mannopyranosyl-chitobiose phosphorylase (UhgbMP; Uhgb_MP)	5-fold β-propeller	Hexamer	[195]
GH149	/	Uncultured bacterium	6HQ6[B], 6HQ8[A,B], 6HQ6[A]	β-1,3-oligoglucan phosphorylase (Pro_7066)	(α/α)6	Dimer	[114]
GT35	AAM52219.1	*Corynebacterium callunae*	2C4M[A,B,C,D]	Glycogen phosphorylase	GT-B	Dimer	Unpublished
GT35	AAC76442.1	*Escherichia coli str. K-12 substr.* MG1655	1AHP[A], 1E4O[A], 1L5V[A], 1L5W[A], 1L6I[A], 1QM5[A], 2ASV[A,B], 2AV6[A,B], 2AW3[A,B], 2AZD[A,B], 2ECP[A	Maltodextrin phosphorylase (MalP; b3417)	GT-B	Dimer	[136,211,212,213]
GT35	AAN59210.1	*Streptococcus mutans* UA159	4L22[A]	Maltodextrin phosphorylase (GlgP; SMU.1564)	GT-B	Monomer	Unpublished
GT35	CAB61943.1	*Arabidopsis thaliana*	4BQE[A,B], 4BQF[A,B], 4BQI[A,B]	α-glucan phosphorylase (PHS2; AtPHS2; At3g46970)	GT-B	Dimer	[127]
GT35	AAB60395.1	*Homo sapiens*	5IKO[A], 5IKP[A]	Glycogen phosphorylase (brain) (bGP)	GT-B	Dimer	[214]
GT35	CAA75517.1	*Homo sapiens*	1EM6[A,B], 1EXV[A,B], 1FA9[A], 1FC0[A,B], 1L5Q[A,B], 1L5R[A,B], 1L5S[A,B], 1L7X[A,B], 1XOI[A,B], 2ATI[A,B], 2QLL[A], 2ZB2[A,B], 3CEH[A,B], 3CEJ[A,B], 3CEM[A,B], 3DD1[A,B], 3DDS[A,B], 3DDW[A,B]	Glycogen phosphorylase (liver)	GT-B	Dimer	[215,216,217,218,219,220,221,222,223]
GT35	AAC17451.1	*Homo sapiens*	1Z8D[A]	Glycogen phosphorylase (muscle) (PigM)	GT-B	Dimer	[224]
GT35	BAK00834.1	*Hordeum vulgare subsp. Vulgare*	5LR8[A,B], 5LRA[A,B]	Plastidial α-1,4-glucan phosphorylase (Pho1; HvPho1)	GT-B	Dimer	[153]
GT35	ACJ76617.1	*Oryctolagus cuniculus*	Dozens available-2GJ4[A] best solution	Glycogen phosphorylase (muscle) (PygM)	GT-B	Dimer	[225]
GT35	AAB68057.1	*Saccharomyces cerevisiae* S288c	1YGP[A,B]	Glycogen phosphorylase (Gph1; YPR160w)	GT-B	Dimer	[226]
GT108	CBZ24449.1	*Leishmania mexicana* MHOM/GT/2001/U1103	6Q50[A]	MTP4 (LMXM_10_1260)	5-fold β-propeller	Monomer	[174]

## Data Availability

Not applicable.

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
