# Peer review of "Discovery and Biotechnological Exploitation of Glycoside-Phosphorylases"

_ijms, 2022, doi:10.3390/ijms23063043_

Round 1

Reviewer 1 Report

Good morning dear authors,

At the beginning, I have to state that your review article is well done. This is evidenced by the presence of your work listed in the references. However, I miss the introduction of the basic kinetic parameters of glycoside-phos-2 phorylases. Furthermore, several formal deficiencies were found, as shown below.

Typos

For organic compound names having N-, the letter N should be written in italics, also in Tables.

Line 688, insert the internet link into References.

Lines 183, 533, there are no spaces between the words.

The numbers denoted by the order of the number of cells should be given in the upper index, for example 103.

Figures

Figure 1 should be divided into individual parts to be shown reaction mechanisms more legible for readers. The quality (contrast) of Figure 2 should be improved. In Figure 3, borders in the case of the last cell is incomplete.

Tables

Tables formatting should be unified. The list of references should be mention in the in the last column on the right. Inner lines should be deleted, where the lines remain only for the table labels and the last line. Remove the inner and outer marks from all lines. Please, follow Instructions for authors.

References

Unify reference format. Use official abbreviations of scientific journals. Add doi numbers everywhere. Delete capital letters in article titles. They will only be at the beginning of the title.

I believe that my comments help you to improve your review article.

With the best regards!

Your reviewer

Reviewer 2 Report

The review by Li et al. concerns the exploitation of glycoside-phosphorylases in biotechnology and encompasses tools for GPs discovery and characterisation. The manuscript is well written and addresses the scope. On top of that the literature work is complete and a precious tool for anyone interested in getting into the topic, or even as a fast source of literature for the experts. Just minor points require attention and must be addressed. Check the text for double spacing and cut the longer sentences into shorter ones. Some passages, especially in the Introduction are too long. Provide abbreviations legend at the beginning of the text as some terms (like CAZymes) are not explicated in the text.

Line 18: in vitro and in cellulo, not in-vitro and in-cellulo

Line 47: I don’t consider appropriate to state that one kind of enzyme is more interesting than the other, they retain different pros and cons

Line 65: α-D-glucose-1-phosphate

Line 90: missing reference

Lines 99-102: the sentence is unclear

Lines 91-102: a figure would help the reader appreciate the difference between GHs and GPs

Line 108: missing reference

Lines 103-120: mention Figure 1 in the text and move it right after this paragraph

Line 190: 4-O-β-D-glucopyranosyl-D-gluconic acid, and in general double check the whole text

Line 200: remove “with preference for galacto-N-biose” as it is clear from the enzyme name

Line 201: remove “with preference for lacto-N-biose” as it is clear from the enzyme name

Line 231: replace GlcUA with the current nomenclature GlcA

Line 238: is it correct for GH94 to be mentioned here?

Line 275: What do the authors mean with “There is no correlation between 3D-folds and GPs mechanism or substrate specificity”? Are the mechanisms and binding modes not retained between members of a family?

Before/after Table 2: a panel figure with the different protein folds and relative substrate sites would be very informative and complement Table 2

Line 234: in cellulo

Figure 2: again, that is not how in vitro and in cellulo are written

Line 427: move here Figure 3

Lines 571-572: should “0.3.106 to 30.106 genes” be “0.3 * 106 to 30 * 106 genes”?

Line 614: have high structural similarities

Line 651: Bacteroidota is the current nomenclature

Line 723: change “as can testify the many interests” to “as testified by the huge interest…”

Line 729: to onto

Line 738 in-vitro

Line 743: in-cellulo
